



# Updating induced seismic hazard assessments during hydraulic stimulation experiments in underground laboratories: workflow and limitations

*Valentin Samuel Gischig[1,2]\*, Antonio Pio Rinaldi[2], Andres Alcolea[3], Falko Bethman[3], Marco Broccardo[4], Kai Bröker[5,6], Raymi Castilla[3], Federico Ciardo[7], Victor Clasen Repollés[2], Virginie Durand[8], Nima Gholizadeh Doonechaly[5,6], Marian Hertrich[5], Rebecca Hochreutener[5], Philipp Kästli[2], Dimitrios Karvounis[3], Xiaodong Ma[9], Men-Andrin Meier[2], Peter Meier[3], Maria Mesimeri[2], Arnaud Mignan[10], Anne Obermann[2], Katrin Plenkers[11], Martina Rosskopf[2], Francisco Serbeto[3], Paul Selvadurai[5], Alexis Shakas[5], Linus Villiger[2], Quinn Wenning[1,5], Alba Zappone[5], Jordan Aaron[1], Hansruedi Maurer[5], Domenico Giardini[5]*

*\*Corresponding author: valentin.gischig@eaps.ethz.ch*
[1]*Geological Institute, Department of Earth Sciences, ETH Zürich, Switzerland*
[2]*Swiss Seismological Service, ETH Zürich, Switzerland*
[3]*GeoenergieSuisse AG, Zürich, Switzerland*
[4]*Dept. of Civil, Environmental and Mechanical Eng, University of Trento, Italy*
[5]*Institute of Geophysical, Department of Earth Sciences, ETH Zürich, Switzerland*
[6]*Center for Hydrogeology and Geothermics, University of Neuchâtel, Switzerland.*
[7]*Department Civil and Environmental Engineering, Northwestern University, USA.*
[8]*GeoAzur, Université Côte d'Azur, France*
[9]*School of Earth and Space Sciences, University of Science and Technology of China, Hefei, China*
[10]*Mignan Risk Analytics GmbH, Switzerland*
[11]*GMuG Gesellschaft für Materialprüfung und Geophysik, Germany*

**Abstract**: Advancing technologies to harvest deep geothermal energy has seen backlashes related to unacceptable levels of induced seismic hazard during hydraulic stimulations. A thorough analysis of induced seismic hazard before these operations has recently become standard practice in the last decade. Additionally, more process understanding of the underlying causes of induced seismicity as well as novel approaches to develop geomechanical reservoirs are being explored in controlled underground laboratory experiments world-wide. Here, we present a probabilistic analysis of the seismic hazard induced by the ongoing hectometer scale stimulation experiments at the Bedretto Underground Laboratory for Geoenergies and Geosciences (BULGG). Our workflow allows for fast updates of the hazard computation as soon as new site-specific information on the seismogenic response (expressed primarily by the feedback afb-value and the Gutenberg Richter b-value) and ground motion models (GMM) become available. We present a sequence of hazard analyses corresponding to different project stages at the BULGG. These reveal the large uncertainty in a priori hazard estimations that only reduce once site-specific GMMs and information on the seismic response of specific stimulation stages are considered. The sources of uncertainty are 1) the large variability in the seismogenic response recorded across all stimulation case studies, as well as 2) uncertain GMMs on the underground laboratory scale. One implication for large-scale hydraulic stimulations is that hazard computation must be updated at different project stages. Additionally, stimulations have to be closely accompanied by a mitigation scheme, ideally in the form of an adaptive traffic light system (ATLS), which reassesses seismic hazard in near-real-time. Our study also shows that the observed seismogenic responses in underground laboratories differ from large-scale stimulations at greater depth in that the seismogenic response is substantially more variable and tends to be weaker. Reasons may be lower stress levels, but also smaller injected volumes accessing a more limited fracture network than large-scale stimulations. Exploring the physical reasons leading to the weaker seismogenic response may reveal ways for safer exploitation



of geoenergy resources. Controlled underground laboratory experiments can readily contribute to this, and – as shown in the presented analysis – are likely to be safe in terms of induced seismic hazard.

## 1. Introduction

Induced seismicity is well known to occur in various underground engineering operations (Kivi et al. 2023) such as hydrofracturing for unconventional gas extraction (Schulz et al., 2020a,b), wastewater disposal from hydrofracturing (Ellsworth, 2013), conventional gas extraction (van Thienen-Visser and Breunese, 2015), $CO_2$ storage (IEAGHG, 2022; White and Foxall, 2016), mining (Lasocki and Orlecka-Sikora, 2008, Wesseloo, 2018) and geothermal projects (Buijze et al 2020). Felt or even damaging induced seismic events have led to halting of various projects (e.g. Basel, Häring et al., 2008; St. Gallen;

Diehl et al., 2019; Pohang, GSK, 2019; Blackpool, UK, Kettlety et al. 2021; Vendenheim; Schmittbuhl et al. 2021) and compromised public support for such projects. Induced seismicity is one of the obstacles for the development of new geoenergy technologies (e.g. EGS or $CO_2$ storage) that could potentially contribute to carbon-free energy generation. For geothermal energy projects, Trutnevyte and Wiemer (2017) proposed a semi-quantitative screening approach to assess to what degree induced seismicity may be a concern for a proposed project. Depending on the level of concern, the hazard posed by induced

seismicity is recommended to be analyzed with varying rigor. One rigorous approach follows the concept of probabilistic seismic hazard analysis (PSHA), that has originally been developed for natural earthquakes (Cornell, 1968), and has been adapted for induced earthquakes (Baisch et al., 2009; Mignan et al., 2015; Bommer et al., 2015; Van Elk et al., 2017; Broccardo et al., 2020). A major difficulty of probabilistic induced seismic hazard assessment (PISHA) lies in forecasting induced seismicity a priori (i.e. before the project), because it relies on (statistical or numerical) models with input parameters that are

site-specific (Mignan et al., 2021) and largely unknown before the actual project has begun. Although the underlying physical processes of induced seismicity are reasonably well understood in principle (Grigoli et al., 2017), the actual manifestation of these processes cannot readily be predicted from the properties of the target rock such as rock type, characteristics of the fracture network, mechanical properties of rock mass and fractures, etc. Within the framework of PISHA, this lack of knowledge and all existing uncertainties are characterized quantitatively and transparently through an appropriate

representation of the epistemic uncertainties and aleatory variability (Broccardo et al., 2020).

Given the difficulty in predicting the site-specific seismogenic response to injections, hazard mitigation schemes – usually termed traffic light system (TLS) - are often proposed to accompany deep stimulation operations to avoid unexpectedly high levels of seismicity. The concept of the TLS, initially proposed by Bommer et al., (2006) for the geothermal project Berlìn, El Salvador, has been and is being applied to many injection operations worldwide (e.g. Helsinki, Ader et al., 2020; Pohang,

Hofmann et al., 2018; Blackpool; Huw et al., 2019; Basel, Häring et al., 2008; St Gallen, Diehl et al., 2017; Schultz et al., 2020b). In its original form, it requires thresholds of earthquake magnitude, ground motion and/or public reactions to distinguish different alert levels, each of which is associated with a set of actions (e.g. a reduction of injection rate or halt of the operations) that may mitigate unwanted levels of seismicity. A collection of magnitude-based thresholds for a range of





cases is shown in Figure 1, which expands on the collection by Bosman et al. (2016). The underlying idea is that the maximum

magnitude observed up to a certain point increases with injected volume so that stopping at a lower magnitude earthquake may

effectively avoid larger magnitude earthquakes that are felt or damaging. Choosing these thresholds also requires anticipating

that seismicity not only continues after stopping an injection, but often reaches the maximum magnitude after injection. Verdon

and Bommer (2021) summarize a range of injection-induced seismicity cases worldwide to explore this so-called trailing effect,

and to arrive at the recommendation that injection should be stopped at two magnitude levels below the magnitude that is to

be avoided. While the effectiveness of such TLS is controversial and debated (Baisch et al., . 2019), a deficiency is seen in the

fact that it is merely reactive and based on static thresholds that do not consider new information on seismicity that becomes

available during injection (Huw et al., . 2019; Kiraly-Proag et al., 2016). So-called adaptive traffic light systems (ATLS), as

an alternative to the classic static TLS, are being developed to alleviate these drawbacks (Kiraly-Proag et al., . 2016, 2018,

Mignan et al., 2017). They rely on the ability to forecast seismic hazard in near-real time by considering the incoming

information on the seismogenic response as seismicity is being induced. The time-dependent seismic hazard estimates are cast

in the probabilistic frameworks that are inherent to the aforementioned *a-priori* PISHA.

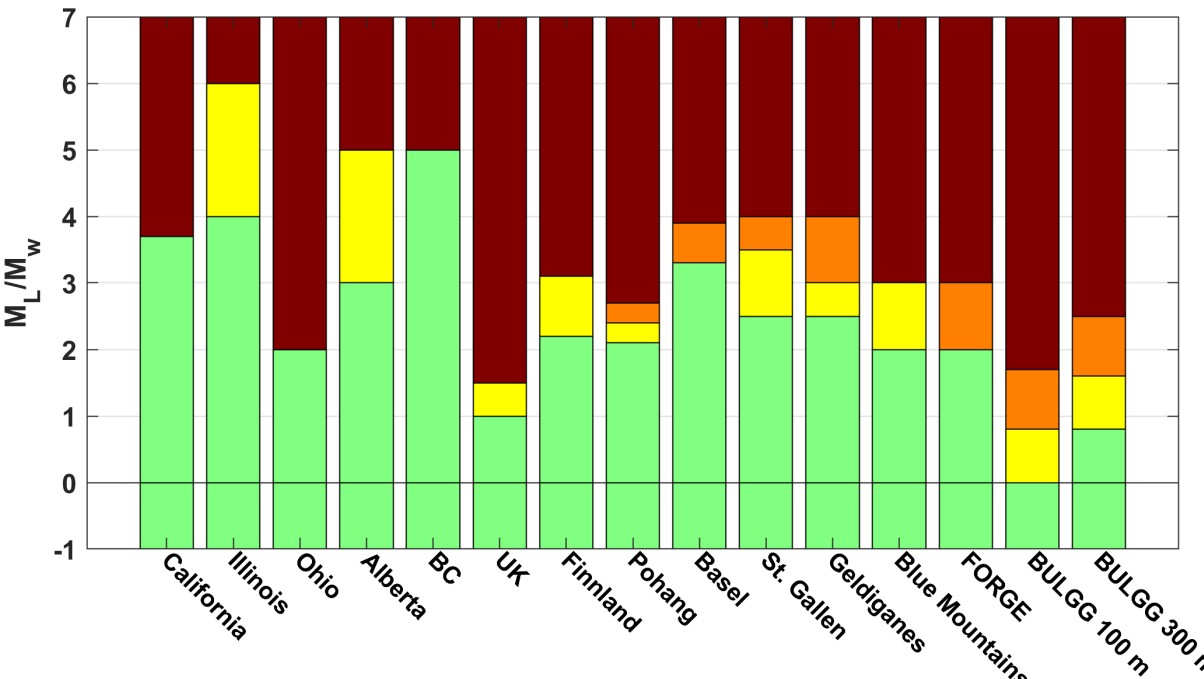

**Figure 1: TLS magnitude thresholds used in various cases expanding on the summary figure by Bosman et al., (2016). California, Illinois, Ohio, Alberta, British Columbia, and the UK are examples of jurisdiction presented by Bosman et al., (2016). Helsinki:**
**Ader et al., (2020); Pohang: Hofmann et al., (2018); Basel: Häring et al., (2008), St. Gallen: Diehl et al., (2017); Geldiganes: Broccardo et al., (2020); Blue Mountains: Norbeck and Latimer, (2024); FORGE: EGI at the University of Utah (2020), BULGG: this study. Note that in some of the cases also ground motion based threshold were used in combination with the magnitude based thresholds. Also, the green, yellow, orange, and red levels do not always imply the same operational consequences. The comparison is made for illustration.**



While technological progress in the field of deep geothermics (and other geoenergy technologies) requires ways to govern induced seismic hazard, research is required to improve our capability of estimating seismic hazard prior to and during reservoir operations, as well as also our understanding of the geomechanical processes during these operations. To this end, great value is seen in down-scaled hydraulic stimulation experiments in underground laboratories. Many projects have been initiated worldwide in the last decade, such as the decameter hydrofracturing experiment in the Aspö underground laboratory in 2015

(Sweden, Zang et al., 2024), the STIMTEC hydraulic stimulation experiment in 2018 (Reiche Zeche, Germany; Boeseet al.,2022), the EGS Collab project in the USA (Sanford Underground Research Facility, USA; Schoenball et al., 2020; Kneafsey et al., 2025), the $CO_2$ injection experiment at Mont Terri, Switzerland (Zappone et al., 2021), the hydraulic stimulation experiments in at the Grimsel Test Site (GTS; Amann et al., 2018), and ongoing hectometer-scale experiments at the Bedretto Underground Laboratory for Geoenergies and Geosciences (BULGG; Ma et al., 2021, Obermann et al., 2024,

Rosskopf et al., 2024, Bröker et al., 2024a; Gholizadeh et al., 2024). The proximity to the stimulated rock volume allows intense multi-parametric monitoring of the stimulation processes based on dense instrumentation (Gischig et al., 2020; Shakas et al., 2020; Plenkers et al., 2023). Since the experiments are conducted at shallower depths and with total injected volume several orders of magnitude lower than for full-scale stimulations, the experimental conditions are not only more accessible and controllable, but likely also safer regarding induced seismic risk. Nevertheless, the experimental equipment and crew are

only few tens to hundreds of meters away from the perturbed rock volume, and in particular at BULGG larger volume injections into an extended fracture network were performed. Thus, it was necessary that for the experiment at GTS and BULGG a seismic hazard analysis be conducted similarly as for the full-scale experiments (Gischig et al., 2016, 2019). However, the goal of these studies is not only to address the actual hazard to people and infrastructure, but also to demonstrate to the public that hazard and risk analysis are an integral part of any stimulation project as much as it is stringent to full-scale stimulations

at great depths. At the same time, the studies serve as a testbed for building and refining PISHA frameworks, in which difficulties and deficiencies can be identified and open research questions be highlighted.

   With these goals in mind, we present here the methodology, strategies and results of the a priori PISHA study conducted for the BULGG (and GTS) experiments. We also demonstrate a strategy for gradually refining the PISHA study as new site-specific information or from similar underground laboratory experiments becomes available. We address the main sources of

uncertainty and highlight how it can be reduced in a systematic, objective way once more site-specific or even interval-specific information is used. We describe knowledge and research gaps that must be filled to improve our capability to predict induced seismic hazard and risk at the 10 – 100 m laboratory scale, as well as on the scale of commercial projects. Thus, while rigorous PISHA has been conducted for mining-induced seismicity (Wesseloo, 2018), gas fields (TNO, 2020) and geothermal projects (EGI at the University of Utah, 2020; Broccardo et al., 2020), etc. we present what is to our knowledge the first PISHA for

hydraulic stimulations in underground laboratories.



## 2.    The Bedretto Underground Laboratory for Geosciences and Geoenergies (BULGG)

The BULGG is in the Bedretto Tunnel in the Swiss Central Alps, which is a 5218 m long adit that connects the Furka railway tunnel with the Bedretto Valley (Figure 2). Since construction in 1982, the Bedretto tunnel remained unlined and unpaved and was mostly used for ventilating and draining the Furka tunnel. In 2018, the Bedretto tunnel has been made available by its

owner (the railway operator "Matterhorn Gotthard Bahnen") to ETH Zürich to conduct research related to geoenergy and other geoscientific topics (Ma et al., 2022). The tunnel runs from NW to SE at an elevation of 1505 m a.s.l. at the junction with the Furka tunnel to 1480 m at the southern portal. The maximum overburden is ~1593 m at tunnel meter (TM) 3100 measured from the south-east portal. At the laboratory level, which occupies a 100 m long enlarged section of the tunnel at 2000 – 2100 TM, the overburden is about 1000 m. The host rock of the laboratory is a granitic body, the Rotondo granite, which has a

boundary to metamorphic crystalline rock units at TM1138 and reaches beyond the junction to the Furka tunnel (e.g. Lützenkirchen and Löw, 2011, Figure 2). The Rotondo granite exhibits subvertical, NE–SW striking, weakly developed foliation as well as SW-NE trending vertical ductile shear zones (Ceccato et al., 2024; Ma et al., 2022; Lützenkirchen, 2002), which often contain fault cores with gouge and cataclasites.

The tectonic seismic hazard in the BULGG region is generally low to moderate (SUIhaz2015, Wiemer et al., 2016). The

regional stress field around Bedretto, as estimated from focal mechanism solutions by Kastrup et al., (2004), is a transitional regime from strike-slip (predominant in the northern Alps and the foreland) to normal faulting (predominant in southern parts of the Swiss Alps). Local stress characterization based on hydrofracturing between TM1750 and TM2250 (Bröker and Ma, 2022; Bröker et al., 2024b) confirms that the overburden stress is close to a principal stress direction ($S_V$ ~25.7 MPa). The inferred maximum horizontal stress direction ($S_{Hmax}$) is approximately WNW-ESE. The estimated minimum horizontal stress

magnitude ($S_{hmin}$ =14.6 ± 1.4 MPa) and maximum horizontal stress magnitude ($S_{Hmax}$ =24.6 ± 2.6 MPa) support that the stress state in the vicinity of the Bedretto Lab is transitional between normal and strike-slip faulting conditions ($S_V \geq S_{Hmax} > S_{hmin}$). The static pore pressure of 2.0 - 5.6 MPa estimated in the stress measurement boreholes is below hydrostatic (maximum 9.8 MPa) implying that topographic effects as well as considerable tunnel drainage and pressure drawdown over the last 40 years have an effect on pore pressure.



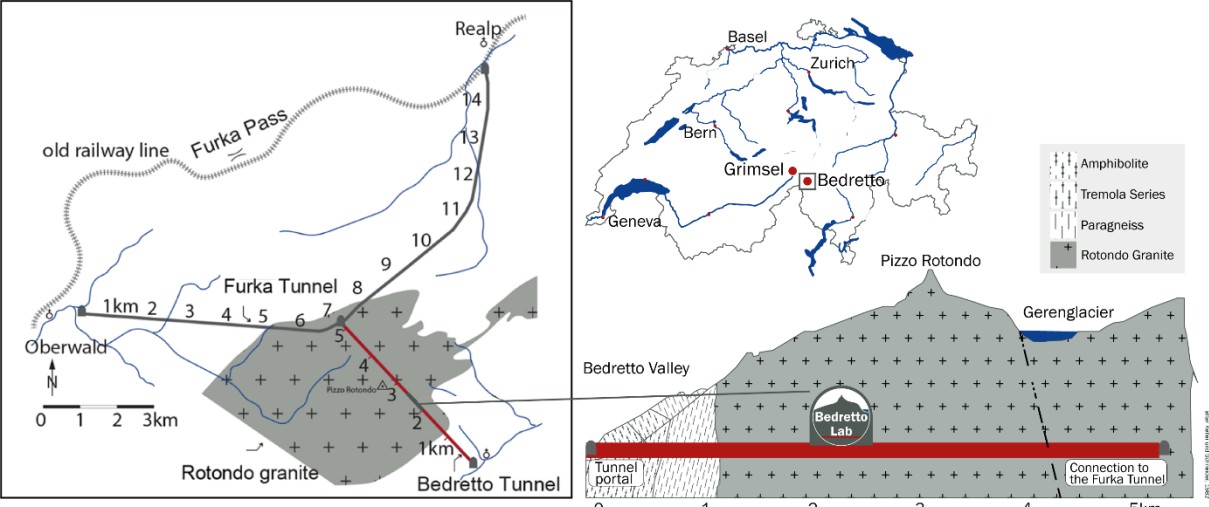

**Figure 2: Overview map and geological cross-section of the BULGG (adopted from Ma et al., . 2022).**

## 3. Instrumentation and experiments between 2020 and 2024

Experimental work in the BULGG between 2020 and 2024 included three projects related to geoenergy: VALTER, DESTRESS and ZoDrEx (Giardini et al., 2022; Meier and Christe, 2023). The goal of the VALTER and DESTRESS projects was to create a reservoir in crystalline rock so that geothermal energy can be extracted or stored by fluid circulation with a minimum induced seismic risk. In the project ZoDrEx, multi-stage stimulations using zonal isolation, innovative drilling, and completion methods were tested. The hydraulic stimulations part of the project DESTRESS was monitored with a network of borehole seismic sensors chains, while those of the project VALTER were accompanied by a multi-sensor monitoring system (Plenkers et al., 2023, Obermann et al., . 2024) that included diverse sensors networks, which allowed inferring details of seismicity, deformation and pressure propagation during, before and after stimulation and helps understanding the seismo-hydro-mechanical responses.

All boreholes drilled for these purposes have been characterized in detail using acoustic and optical televiewer logs, borehole georadar, borehole geometry logs (caliper, deviation), fluid electrical conductivity and temperature logs and/or spectral gamma logs. Hydraulic characterization within the multipacker systems allowed inferring hydraulic conductivity and connectivity of complex structures and the borehole sections. The results of the characterization campaigns are combined to a geological and rock mechanical model of the target experiment volume (Ma et al., 2022).

The first boreholes CB1, CB2, and CB3 were drilled in September 2019, followed by the first two hydraulic stimulation tests with straddle packers in CB1 in February 2020 (264 – 298. 5 m depth, see Figure 3a, Table A1, Shakas et al., 2020). These involved injection volumes of each about 5 m$^3$.



Later these boreholes were redrilled to enlarge diameter and to transform them into monitoring boreholes and renamed MB1, MB2, and MB3, respectively. In May and June 2020, the injection/production boreholes ST1 and ST2 as well as MB4 were drilled. After instrumentation of MB1 – MB4 between February and July 2020 (Plenkers et al., 2023; Golizadeh et al., 2024), the hydraulic stimulation experiments of the project DESTRESS were conducted in the lower parts of ST2 (5 intervals between

306-345 m depth in November 2020) and of ST1 (7 intervals between 268 – 344 m depth in December 2020). These stimulations were done with hydraulic straddle packers by the company GeoEnergie Suisse (GES).

For most stimulations in ST2 (i.e. for those that a stimulation was possible with the given pump specifications), a test stimulation of about 5 m$^3$ was performed before 10 – 60 m$^3$ were actively injected into the rock during a main stimulation. Such test stimulation (referred to as TS-TLS) was needed to update seismic hazard forecasting models. Thus, the test gave a first

understanding of how hazardous the stimulation of the interval was. This ensured that the main stimulation was safe and allowed testing forecasting capabilities of the models. Then some minor re-stimulation occurred for the uppermost intervals, but with volumes limited to about 10 m$^3$.

For the stimulations in ST1, a more powerful pump was available, which allowed injecting a larger volume of fluid in the given time (between 65 – 160 m3). Because the fracture systems seemed much more permeable in this borehole compared to

ST1, only four out of seven intervals could be pressurized such that seismicity was induced. During these stimulations, no TS-TLS injections were performed.

In early 2021, the borehole ST1 was completed with a multipacker system that allows access to individual intervals using sliding sleeves (Figure 3b; part of project ZoDrEx). In May 2021, hydraulic stimulations were performed by GES in intervals 1+2 (i.e. combined), 4 and 6 of the multipacker system (project VALTER) with pumps allowing injection at several hundreds

of l/min. The bottom part of borehole ST2 (332-345 m) was also stimulated. Additional monitoring boreholes (MB5, MB7 and MB8) were drilled with percussion drilling technique (part of project ZoDrEx) and instrumented in July 2021. As part of the ZoDrEx project, stimulations were conducted in boreholes ST2 to tests notches at various depth.

Finally, between December 2021 and August 2023, further hydraulic stimulations by ETH Zürich were performed in intervals 7 to 14 in ST1 (Obermann et al., 2024) with volumes ranging from 0.36 to 274 m$^3$. These stimulations benefited from the

proximity to the monitoring boreholes that contain a dense network of various types of seismic sensors (Plenker et al., 2023). The stimulation program included two phases. In Phase 1 (November 2021 to March 2022) intervals 7 to 14 were stimulated with a comparable injection protocol using two injection stages of each a few hours. The goal of these injections was to screen the seismic and hydromechanical responses of each interval. In Phase 2 (June 2022 to July 2023), selected intervals were revisited and either stimulated with larger volumes to access a larger rock volume (Interval 8, 9+10) or to test dedicated

injection protocols (Interval 11 and 12) (see Obermann et al., 2024 for further explanation).

Table A2 in the Appendix summarizes the results of all stimulations in terms of injected volume and seismicity characteristics.



**Figure 3:** Borehole configuration at the BULGG. a) Injection and monitoring boreholes, injection intervals and seismicity during the DESTRSS project. Injections were done with a movable straddle packer. b) Injection and monitoring boreholes, intervals and seismicity during the VALTER project. Injections were done in fixed installed packers with sliding sleeves.



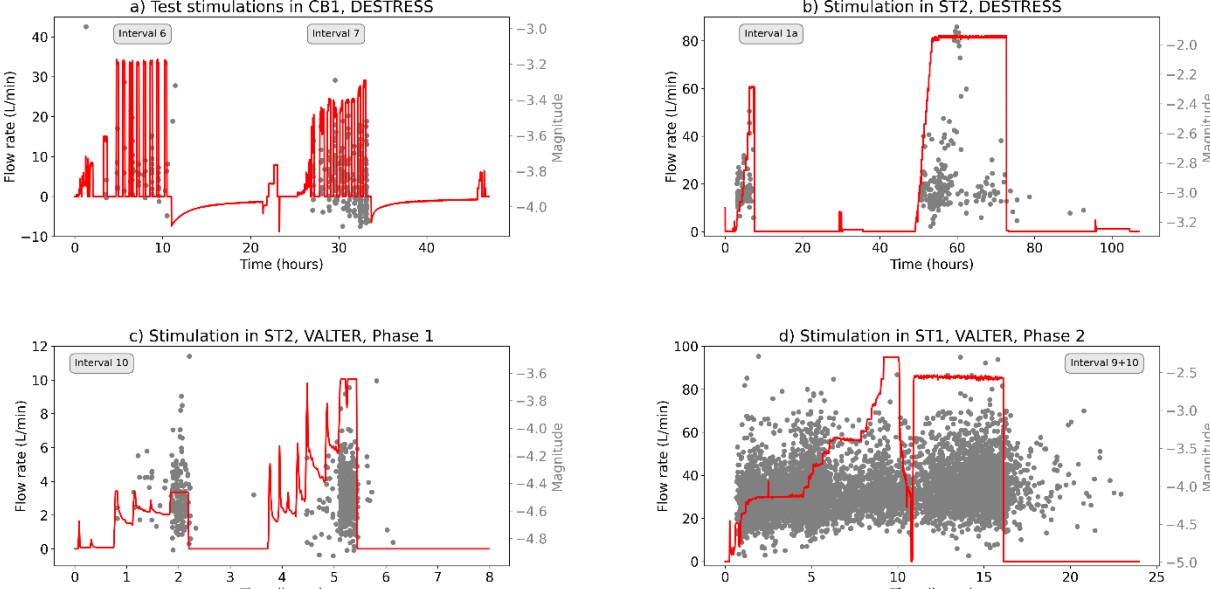

**Figure 4: Examples of hydraulic stimulations of different project phases: a) Initial test Stimulation in CB1 as part of project DESTRESS, b) Example of hydraulic stimulation of ST2 (DESTRESS) with hydraulic packers c) Example of stimulation using fixed packers with sliding sleeves in ST1 as part of Phase 1 project VALTER. d) Example of stimulation in ST1 as part of Phase 2 project VALTER.**

## 4. Sequence of induced seismic hazard studies

As the subject of this article is not only the methodology of computing PISHAs for hydraulic stimulation but also how the PISHA results evolve as new and site-specific information become available, we present the following PISHA studies:

- **Study 1, GTS a priori (state 2016):** Before conducting hydraulic stimulation experiments at the GTS in 2017 (Villiger et al., 2020), a first PISHA was performed using the information on seismogenic responses from different case studies worldwide because no information was available for the underground laboratory scale (Gischig et al., 2016).

- **Study 2, BULGG a priori (state 2019):** A first a PISHA for BULGG has been conducted in May 2019 prior to any injection test and during the construction of the BULGG (Gischig et al., 2019). The analysis could benefit from experience on seismogenic responses from the GTS (Villiger et al., 2021) as well as from Aspö (Kwiatek et al., 2018). However, no site-specific information on BULGG was available.

- **Study 3, BULGG update 1 (state 2021):** After the DESTRESS stimulations in boreholes CB1, ST1 and ST2, the PISHA was updated to include the new information on the seismogenic response in the lower part of the reservoir (Figure 3a). Given the relatively low number of events per stimulation, all seismicity recorded per borehole was combined to compute estimates of the seismogenic response. Note that the uncertainties of seismic locations and





magnitudes are larger than for the shallower part of the volume stimulated during VALTER, because of the larger distance to high-resulution seismic network.

- **Study 4, BULGG update 2 (state 2023):** With the seismogenic responses estimated from VALTER stimulations, which are based on the high-resolution monitoring system, another update of the PISHA was made. The study can be seen as a generic study for the BULGG and allows planning experiments in the same rock volume (e.g. the M-zero experiment performed in April/May 2024 described below), or in other parts of the laboratory for which no site-specific information in available. Given the quality of seismicity catalogues from within the high-resolution part of the seismic monitoring network at shallower depth, the seismogenic responses of each interval individually has been used.

- **Study 5, BULGG M-zero:** In preparation for the so-called M-zero experiment - an extended stimulation experiment with the goal of inducing an Mw0.0 event as part of the earthquake physics project FEAR (e.g. Volpe et al., 2023) – an experiment-specific PISHA was computed. Only parameters from VALTER intervals 8, 9, and 11 were used for this study (highlighted in Figure 7c and d), because they are closest to the target interval 11 and seismicity showed that the same fracture network was activated (Obermann et al., 2024). Additionally, the parameter sets only included stimulations with injected volumes > 5 $m^3$ as they were deemed more representative to the planned M-zero experiment, which was designed to potentially reach up to 100 $m^3$ injected volume.

## 5. Method

Generally, probabilistic seismic hazard analysis (PSHA) requires that a wide range of datasets, models, and methods proposed by the larger technical community to be relevant to the hazard analysis is considered (Cornell, 1968; McGuire and Arabasz, 1990; Bommer and Abrahamson, 2006). PSHA must appropriately represent the uncertainties in the assessment and represent the range of technically defensible interpretations. PSHA does not only consider worst-case scenarios, but all possible outcomes, which allows defining an expected, mean or median outcome. Thus, PISHA (i.e. probabilistic *induced* seismic hazard analysis) itself must not be conservative in choosing the methods, models, or model parameters. Conservatism comes in by choosing an acceptable hazard level in the design (e.g. of buildings, infrastructure etc in case of *natural earthquakes*, or of hydraulic stimulations, traffic lights system, etc in case of *induced earthquakes*) that may be conservative.

Here, we apply PISHA to assess the impact of injection-induced earthquakes during experiments at the BULGG for a range of possible injection volumes and distances. The approach is visualized with the logic tree in Figure 5. The different models and parameter sets used in each logic tree branch represent the epistemic uncertainties. The aleatory variability is considered by assigning uncertainties to the model parameters. Each branch of the logic tree is sampled corresponding to an assigned weight, which has been defined through expert solicitation. Note that the weights vary for the different updates of the hazard computation, as will be explained later.



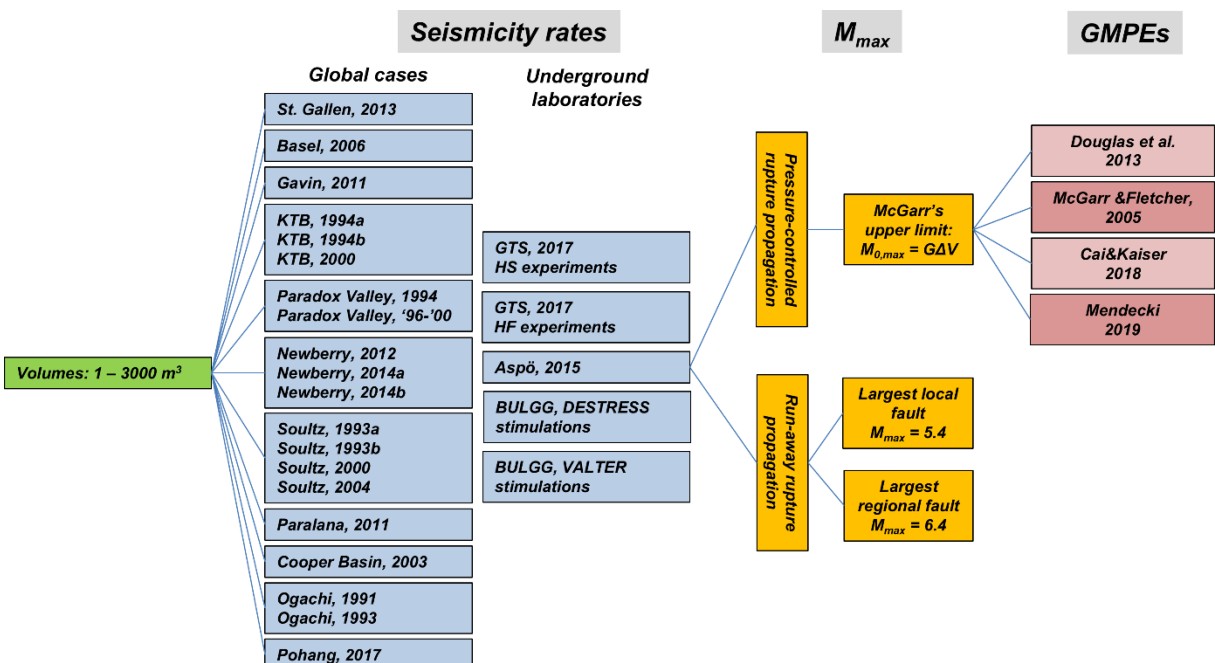

**Figure 5:** Logic tree for the probabilistic induced seismic hazard analyses for the GTS and BULGG stimulation experiments.

*Magnitude rates*

In the first layer of the logic tree (Figure 5), the volume-dependent magnitude rates are estimated. We build on the concept by Shapiro et al., (2010), who proposed a statistical seismicity model that gives an estimate of the cumulative numbers of earthquake N exceeding a magnitude level $M_i$ based on volume V(t) injected up to a time t and a site-specific parameter referred to as seismogenic index. Mignan et al., (2017) refined the seismicity model with an alternative description of the post-shut-in seismicity decay:

$$N_{M \geq Mi} = \begin{cases} 10^{a_{fb}-bM_i}V(t) & t \leq t_{shut-in} \\ 10^{a_{fb}-bM_i}\dot{V}(t_{shut-in})\exp\left(-\frac{t-t_{shut-in}}{\tau}\right) & t > t_{shut-in} \end{cases} \quad (1)$$

Analogous to the seismogenic index, they introduced the activation feedback parameter $a_{fb}$. b is the Gutenberg-Richter b-value and $\tau$ defines the decay of seismicity after a halt of injection (i.e. shut-in of the borehole). A catalogue of estimates from different cases are given by Mignan et al. (2021) In our case, an estimate of $\tau$ is not available for all considered case studies. Since we do not need to model the temporal decay of seismicity explicitly, it is sufficient to use the fraction of events that occurred after shut-in of the total number of events to account for the post-shut-in trailing effect. The approach relies on the simplifying assumption that the b-value remains constant during injection and after shut-in. We here use 47 parameter sets of injections at 14 stimulation cases worldwide, at the GTS and the BULGG (Table A1). In case an estimate $a_{fb}$ is not available, we used the seismogenic index $\Sigma$ reported in the references. Note that for cases, for which a standard deviation of the b-value



was not available, we used a nominal value of 0.05. The error of $a_{fb}$ depends on the error of the b-value; thus, for different realizations of the b-value a corresponding $a_{fb}$-value was computed. For cases, for which the percentage of events after shut-
in was not available, we used a nominal value of 10%.

In our sequence of hazard computation updates, the weighting of the parameter sets in Table 1 constitutes the main adjustment in the hazard estimates between each update (besides GMMs, see below). The weighting was determined based on an *expert elicitation*, in which scientists compare the similarity of each case study with the conditions at the BULGG in terms of rock type, depth, stress level and regime, injected volume and the process of inducing seismicity. Additionally, the reliability of
each parameter set based on the underlying magnitude estimates is rated. The numerical ratings are evaluated to arrive at a weight for each case study (Figure 6). The weights of all three scientists are averaged. These correspond to the weights for the BULGG update 2.

In the sequence of our five hazard estimates the weights were adjusted (Figure 6b). For the Grimsel experiments at the GTS, no parameters on the seismogenic response to injection were available for underground laboratories. Similarly, the parameters
of Pohang were not available. Hence, we had to solely rely on the other worldwide sets. (Note that this differs from the original GTS hazard study by Gischig et al., (2016), in which each parameter set received equal weight. The weights were adjusted here to conform to the later hazard computations for better comparability.). The stimulation experiments at GTS and Aspö were conducted between 2015 and 2017. Thus, these datasets were included in the a priori hazard computation for BULGG. In the update before the VALTER stimulation starting in November 2021, the DESTRESS stimulations became available as
well as information on the Pohang stimulations. Figure 6b illustrates how the weights for case studies outside of the BULGG receive step-wise smaller weights as underground laboratory experience or even site-specific experience becomes available. The parameter sets in Table 1 are shown in Figure 7 together with an illustration of how the $a_{fb}$-/b-value field is sampled in the different hazard computations.

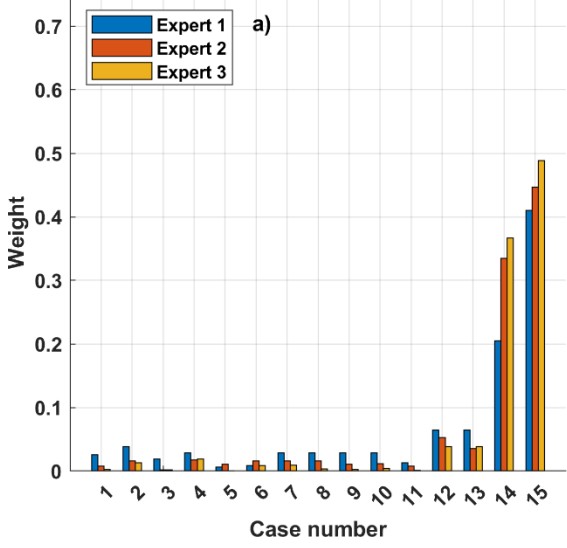

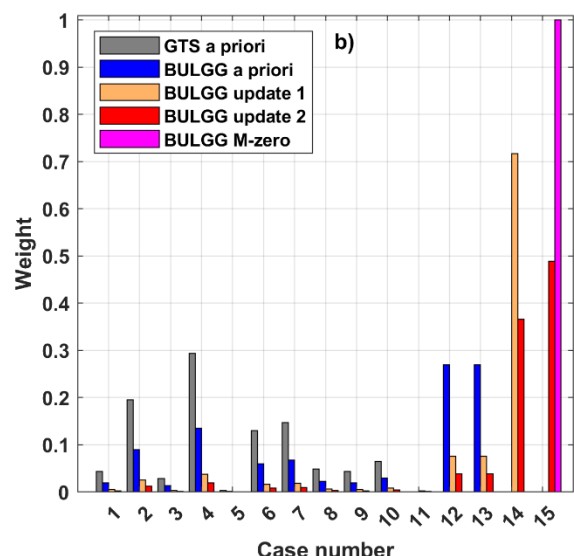





**Figure 6: a) Weighting of case studies derived by comparing each case study with the conditions at the BULGG by four scientists. b) Weights used for each update of the sequence of hazard computations.**



**Figure 7: a) $a_{fb}$- and b-values used considered in the PISHA. b) Probability density function (PDF, grey scale) of the $a_{fb}$- and b-values
chosen in the random realizations of the PISHA for the example of the BULGG update 2 analysis. c) $a_{fb}$-values in
relationship to injected volume. d) b-values in relationship to injected volume.**

*Maximum possible magnitude*

Equation 1 predicts a finite seismicity rate even for large unphysical magnitudes. Thus, the frequency magnitude distributions
(FMDs) are truncated at a maximum magnitude that can possibly occur based on physical or statistical/empirical
considerations. This maximum possible magnitude $M_{max}$ describes very extreme and rare events, i.e. the tail of a distribution



and is difficult to assess and typically very uncertain. As shown by e.g. Mignan et al. (2015) or Bommer and Verdon, (2024), the choice of $M_{max}$ has a small impact on hazard and risk for commercial scale projects, at typical recurrence times, because these events are very rare and risk is dominated by moderate events.

For the case of induced seismicity, the choice of maximum possible magnitude follows two different viewpoints that are discussed in the literature: 1) Some authors (e.g. McGarr, 2014) argue that there is a fixed upper threshold for a physically maximum possible magnitude that can be induced by fluid injection. The magnitude can be computed from the scalar seismic moment $M_0 = G\Sigma V$, where G is the shear modulus of the medium (here G=20 GPa) and V is the total injected volume. Nonetheless, McGarr (2014) argues that larger magnitudes cannot be entirely excluded due to the uncertainty in the analysis

and because a different triggering mechanism in addition to fluid injection may contribute. 2) Other authors (Atkinson et al., 2016; Eaton and Igonin, 2017) argue that Mmax is the same as for tectonic earthquakes. Thus, the FMDs can be extrapolated towards large magnitudes representing earthquakes that would occur if the largest fault in the region would rupture entirely. This view point is supported by the recent hydraulic stimulation in Pohang, South Korea, which has likely induced a Mw5.5 (Grigoli et al., 2018, GSK, 2019). For the case of Pohang, McGarr's estimated maximum possible magnitude for the injected

volume of ca. 10'000 m$^3$ was Mw3.7 (Figure 8).

A numerical analysis by Gischig (2015) using coupled rate-and-state frictional behavior and hydromechanics (McClure and Horne, 2011) showed that a critically-stressed fault (i.e. a fault verging on failure) may indeed rupture beyond the pressurized fault area and become an earthquake as large as a tectonic one (so-called *run-away rupture propagation*). However, if the fault is not critically-stressed (e.g. not-optimally oriented in the stress field), then rupture arrests at the pressure front (*pressure-*

*controlled rupture propagation*). The former case implies a maximum possible magnitude corresponding to the tectonic one, while the latter implies that an upper threshold as suggested by McGarr (2014) is feasible. These outcomes confirm the results of slip-weakening fault models by Garagash and Germanovic (2012), who similarly distinguish between these two rupture propagation regimes. Recently, Ciardo and Rinaldi (2022) demonstrated that the ramp-up of the pressurization may also have an important role in determining the maximum magnitude, but again confirmed that for critically stress fault a run-away rupture

can occur. Recent statistical analyses show that the maximum magnitude can be bound or unbound (Schultz, 2024)

The effective stress level, that may play a role in how likely run-away ruptures occur, increases to a first order linearly with depth. It is thus plausible that injections at shallower depth trigger a different seismic response than at greater depths, which is also evident from the dependency of a-value and b-value of tectonic events on faulting style and depth (e.g., Spada et al., 2013; Petruccelli et al., 2019). Likely, the depth-dependence of the $a_{fb}$-values, b-value and run-away rupture probabilities are coupled,

yet limited data exist to define the dependencies.

In our view, the assumption that run-away ruptures are less likely at shallower depth is well captured by the $M_{max}$ branch based on McGarr's limit. Further, in the case of run-away ruptures, we consider two fault sizes. Thus, the epistemic uncertainty in the assessment of the maximum possible magnitude $M_{max}$ is computed as follows:





- $M_{max}$ = 6.4 represents the mean maximum tectonically possible magnitude in the Swiss Alps following the national
Swiss hazard assessment of 2015 (Wiemer et al., 2016). This would represent the case, where a rupture is triggered
on an unknown and critically pre-stressed large fault that extends into the basement (weight 25%).

- $M_{max}$ is defined by the largest fault in the region around the BULGG. In a study of brittle fault zones within the
Gotthard Massiv, Lützenkirchen (2002) maps a fault that intersects the Rotondo Granite at about 2 km distance from
the lab. The length of the fault is mapped with 7km. In this scenario, it is considered possible, that an injection finds
a pathway to the fault and can trigger the entire fault with a rupture area of 7x7 km. A calculation with a stress drop
of 3 MPa results in $M_{max}$ =5.4 (weight 25%).

- $M_{max}$ is a function of injection volume following McGarr, (2014). (Weight 50%).

We consider $M_{max}$ as a random variable reflecting further epistemic uncertainty, i.e. our limited knowledge in the given exact
upper-bound. We consider a Gaussian distribution with a standard deviation of 0.3 (for McGarr's $M_{max}$) and 0.8 (for the tectonic
$M_{max}$'s). Figure 8 shows McGarr's relationship along with maximum observed magnitudes from case studies from various
injection operations. Injections of 1 m$^3$ or 1000 m$^3$ correspond to $M_{max}$ of M1.0 and M3.0, respectively.

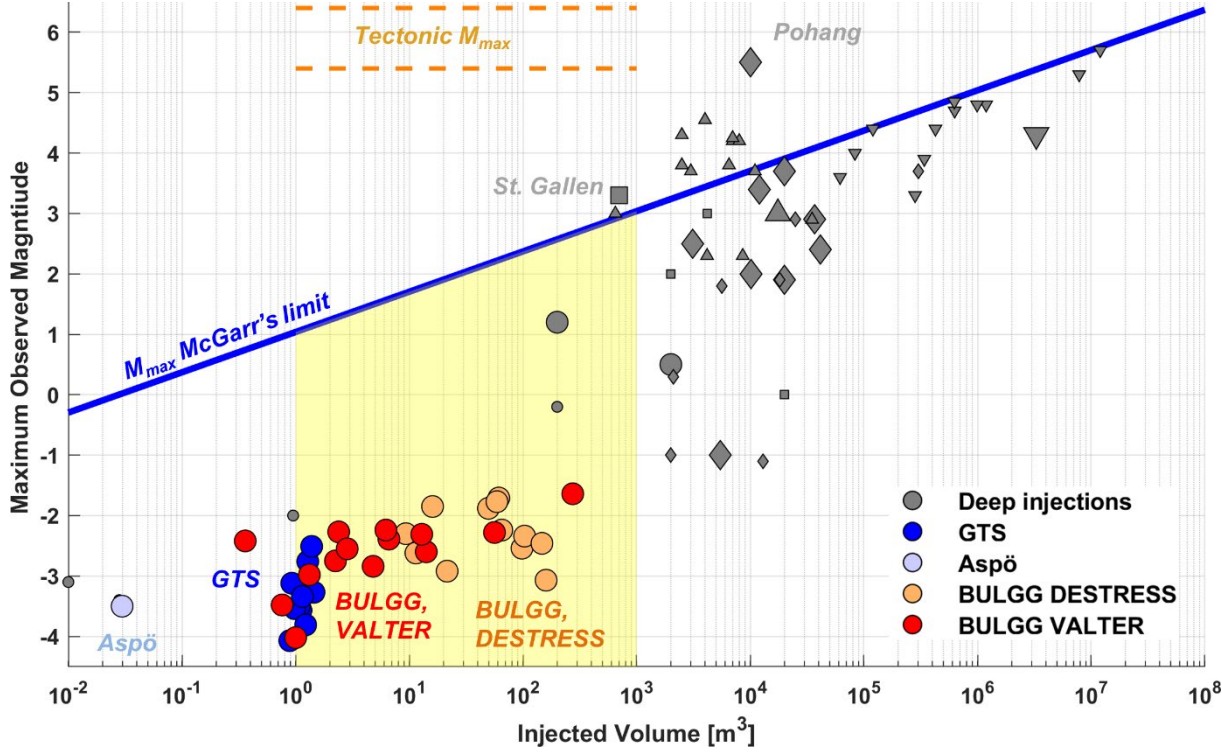

**Figure 8: Different versions of the maximum possible magnitude $M_{max}$ for the volume range considered in our study (1-1000 m$^3$).**
**Also shown are the observed maximum magnitudes during various injections (Villiger et al., 2020; Obermann et al., 2024). Large**
**markers indicate case studies for which $a_{fb}$– and b-values are available to our analyses. Note that the limits for the maximum expected**
**magnitude proposed by van der Elst et al. (2016), as well as the limit for the maximum possible magnitude by Galis et al. (2017) were**
**not included in the Figure as they were not used to defined $M_{max}$ in this study.**



### *Ground Motion Models*

In the third layer of the logic tree, ground motion models (GMM) have to be used to estimate actual ground motion (i.e. peak
ground velocity, PGV) at a given distance R from the earthquake for an earthquake of magnitude Mw. Due to the short
distances and the presumably small magnitudes in our case, we cannot use GMMs for tectonic earthquakes, which would be
widely available in the literature. However, attempts have been made to derive GMMs for induced seismicity in underground
mines or for geothermal sites. Butler and Aswegen (1993) report GMMs from underground mines that depend on a local
magnitude $M_L$ (range $M_L$ = 0.5 – 4.0, R =150 – 10'000 m). Similarly, Hedley, (1990) reports $M_L$-based GMMs from
underground mines. The equation by McGarr and Fletcher, (2005) from mining-induced seismicity is expressed in terms of
seismic moment and $M_w$ (range $M_w$>1.0, R = 500 – 10'000 m). Cai and Kaiser (2018) propose to use equations that have the
same functional form as the one reported by McGarr (1984) and give a possible range of constants derived from many case
studies. The equations proposed by Mendecki (2019) differ in the functional form and in that potency is used instead of the
seismic moment (range Mw>0.2, R = 50 – 500 m). A GMM specifically for induced seismicity in the context of deep
geothermal was proposed by Douglas et al., (2013) (range Mw>1.0, R=1500 – 50'000 m)

There is a consensus in these studies that GMMs must be derived from site-specific seismic data despite similarities in the
functional form between sites (e.g. Cai and Kaiser, 2018; Mendecki, 2019). In our case, local seismicity data was not available
before for the a priori analysis for GTS and BULGG. However, seismicity data became available once hydraulic stimulation
started at the BULGG (Obermann et al., 2024; Rosskopf et al., 2024; Mesimeri et al., 2024). Seismicity induced by hydraulic
stimulations was recorded by a high-resolution seismic network based on highly sensitive acoustic emission sensors,
accelerometers, and borehole geophones (Plenkers et al. 2023). Waveforms recorded with the accelerometers and geophones
provide estimates of PGV for induced earthquakes. In addition, seismic stations in the tunnel and on the ground surface as well
as the borehole geophones recorded natural seismicity regional to the BULGG (Mesimeri et al., 2024). Using values of PGV
from a distance of 3 km around the BULGG, we can assess, which of the ground motions best fit the local observations (Figure
385  9).

Thus, for our PISHA sequence, we chose the following GMMs from literature:

- For the *a priori GTS and a priori BULGG analyses*, we chose the GMMs by McGarr and Fletcher (2005, Eq. 3
therein), Cai and Kaiser (2018, Eq. 2-2, p.56), Mendecki (2019, Eq. 6 therein assuming shear modulus G=20 GPa to
translate potency to seismic moment), as well as Douglas et al., 2013, Table 2 based on corrected data therein). We
did not consider the equations by Butler and Aswegen (1993) and Hedley (1990) because they rely on $M_L$ and a
conversion to $M_w$ has not been derived for these data sets and using other reported conversion equations (e.g.
Deichmann, 2017; Edwards et al., 2015) would introduced further uncertainty. The four chosen GMMs were equally
weighted (i.e. 25% each) to account for the epistemic uncertainty.



- • For the *BULGG update 1 and 2* and *BULGG M-zero,* we chose equations by McGarr and Fletcher, (2005) and
Mendecki (2019) with equal weight (50% each), because they fit the observed PGVs best (Figure 9b-e). We discarded
the equations by Cai and Kaiser (2018) and Douglas et al., (2013) that systematically deviated from the observations.

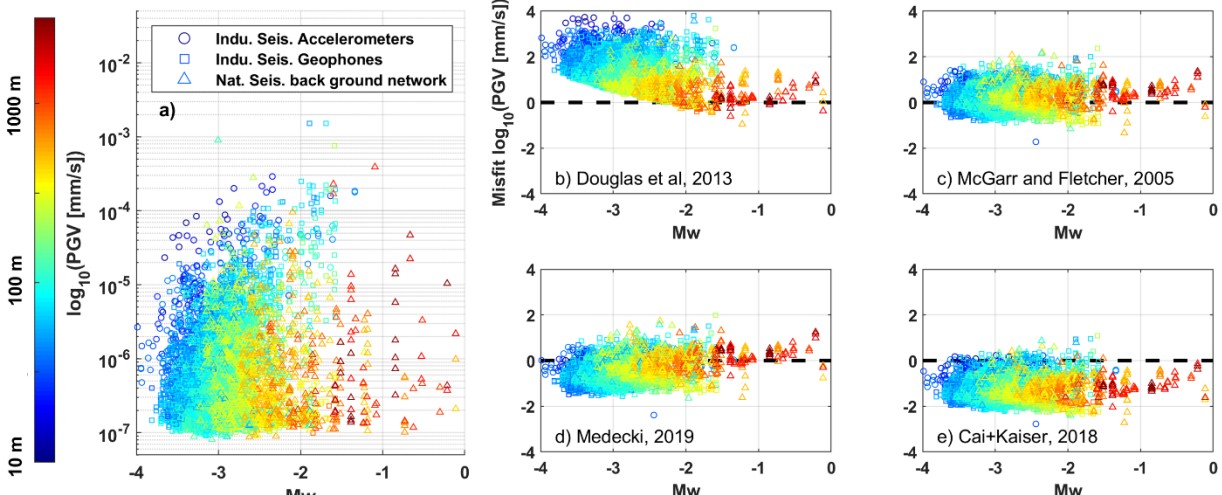

**Figure 9: a) PGVs observed within and around the BULGG with the high-resolution monitoring network and the background**
network (Mesimeri et al., 2024). Ground motions recorded at accelerometers, borehole geophones as well as tunnel and surface
seismic stations were used. b-e) Comparison of the GMMs by McGarr and Fletcher (2005), Cai and Kaiser (2018), Mendecki (2019),
as well as Douglas et al. (2013) motion data that is available for these distances.

Figure 10 shows the probability density function (PDF, grey shading) of the PGV as a function of magnitude at distance 150
m (representative distance of the BULGG cavern to stimulation experiments) and 2000 m (minimum distance to infrastructure
at the tunnel portal as well as the Furka tunnel railway infrastructure). For this, the chosen GMMs were sampled randomly $10^5$
times using the corresponding weights (epistemic uncertainties) and respective uncertainty of each equation (aleatoric
uncertainty). If all four GMMs are combined (as done for the a priori GTS and BULGG analyses), the 10 and 90% percentiles
from this distribution cover more than two orders of magnitude. For example, at 150 m distance, a PGV of 30 mm/s is exceeded
with an event of magnitude of about Mw2.3, but with a range from Mw1.4-3.8 (Figure 10a). At 2000 m distance, the magnitude
to exceed a PGV of 30 mm/s is 4.0, but with a range from Mw3.0-5.0. However, once site-specific information on ground
motions is considered, uncertainties reduce substantially. At 150 m distance, 30 mm/s are exceeded for Mw2.4 (median) with
a range of $M_w2.0 - 2.7$ (10 and 90% percentiles)





**Figure 10: a) PGV at a distance of 150 m (representative distance to injections of VALTER) estimated based on all four GMMs including their uncertainties. The gray shading in the background represent the probability density function. For this, the five equations are weighed equally and are sampled randomly with the corresponding uncertainties in the PGV estimates. b) The same for a distance to the Furka railway tunnel infrastructure.**

*Hazard thresholds*

Induced earthquakes relevant to our context (typically $M_w << 2.0$) have frequencies that are larger than 10 Hz. Thus, methods commonly used in earthquake engineering focusing on large damaging earthquakes have limited applicability. Solutions can be found in mining literature or from norms dealing with vibrations from blasting, construction or traffic. The Swiss Norm SN 640 312a can be used to define thresholds at the tunnel levels. It defines three levels of the excitation frequency, i.e. how often it occurs: occasionally, frequently and permanently. Vibrations from blasting or, as assumed here, from induced earthquakes occur occasionally. Further, the norm distinguishes buildings and infrastructures into four classes of vulnerability (or sensitivity): very low sensitivity, low sensitivity, normal sensitivity, and high sensitivity. Although tunnels and caverns in hard



rock are considered very low sensitive, we prefer to classify the unsupported caverns of the BULGG to be in the class low sensitive and the equipment and machinery as well as railway infrastructure of the Furka tunnel to be normal sensitive.

The threshold values for PGV for frequencies 8 – 30 Hz are 15 mm/s for normal sensitivity and 30 mm/s for low sensitivity.

The norm states that damage becomes likely at values twice these thresholds (i.e. 30 mm/s and 60 mm/s), while severe damage only occurs at a multiple of the values. In the following, we use 30 mm/s as PGV threshold (e.g. Figure 10). These threshold values are in agreement with the observations of damage in mines (Cai and Kaiser, 2018, p81) who describe the following damage classes 1) No damage: PGV<50 mm/s), 2) falls of loose rock: 50<PGV<300 mm/s, 3) falls of ground: 300<PGV<600 mm/s, 4) severe damage: PGV>600 mm/s. The threshold agrees with those discussed in other hazard analyses in the literature:

e.g. Ader et al., (2020) proposed 7.5 mm/s for cosmetic damage to buildings, and 1 mm/s for human perception. Cremen and Werner (2020) use 15 mm/s as the threshold for cosmetic damage to buildings.  Thus, the proposed threshold of 30 mm/s can be considered conservative regarding substantial damage.

### 6.   Results

*Magnitude rates*

Sampling the logic tree (Figure 5) 100'000 times results in the full range of possible outcomes regarding the probability of exceeding a magnitude Mw. Figure 11 shows the multitude of probability curves (represented in grey shading as probability density function, PDF) for an injection volume of 100 m$^3$ for each version of the hazard analysis It is important to note, that we refrain from normalizing the probability to a time-scale (i.e. annualization; Wesseloo, 2018). The probability is understood

as per stimulation experiment, which may typically last a few hours to a few days depending on the experimental design (injection volume, pressure and flow rate) that is a function of interval properties. For comparability with commonly acceptable annualized hazard or risk levels, one would normalize the probability with the duration of the stimulation experiment (e.g. at a typical experimental flow rate of 30 l/min injection rate and a volume of 100 m$^3$ the experiment would last 55 hours, excluding shut-in time).

The range of the curves – also represented by the 10% and 90% percentiles – is comparably narrow for the GTS a priori analysis, for which only parameters of deep injections and no underground laboratories parameters are considered. The outcome may be seen as the outcome of a generic a priori hazard analysis for deep injections. Once underground laboratories are included the range of outcomes spreads, because the b- and $a_{fb}$-values from underground laboratories cover a range with much lower $a_{fb}$-values and higher b-values as the deep injections (Figure 7). The 90% percentile decreases towards smaller

magnitudes for a given probability but to a much lesser degree than the median. The median changes substantially once the BULGG stimulations are available and the given weight is much higher than for all other case studies (Figure 6). For instance, the expected magnitude (i.e. the magnitude that occurs with a rate 1 or the equivalent exceedance probability of 0.63) drops from Mw1.75 (GTS a priori) to Mw-2.0 for the subsequent analyses. For the BULGG M-zero analysis, only the hydraulic





stimulations deemed most representative for Interval 11 are considered. Consequently, the range of hazard estimates collapses

to a narrow range and the expected magnitude (i.e. rate 1, probability 0.63) is Mw-1.3 (range -0.7 to -1.6).

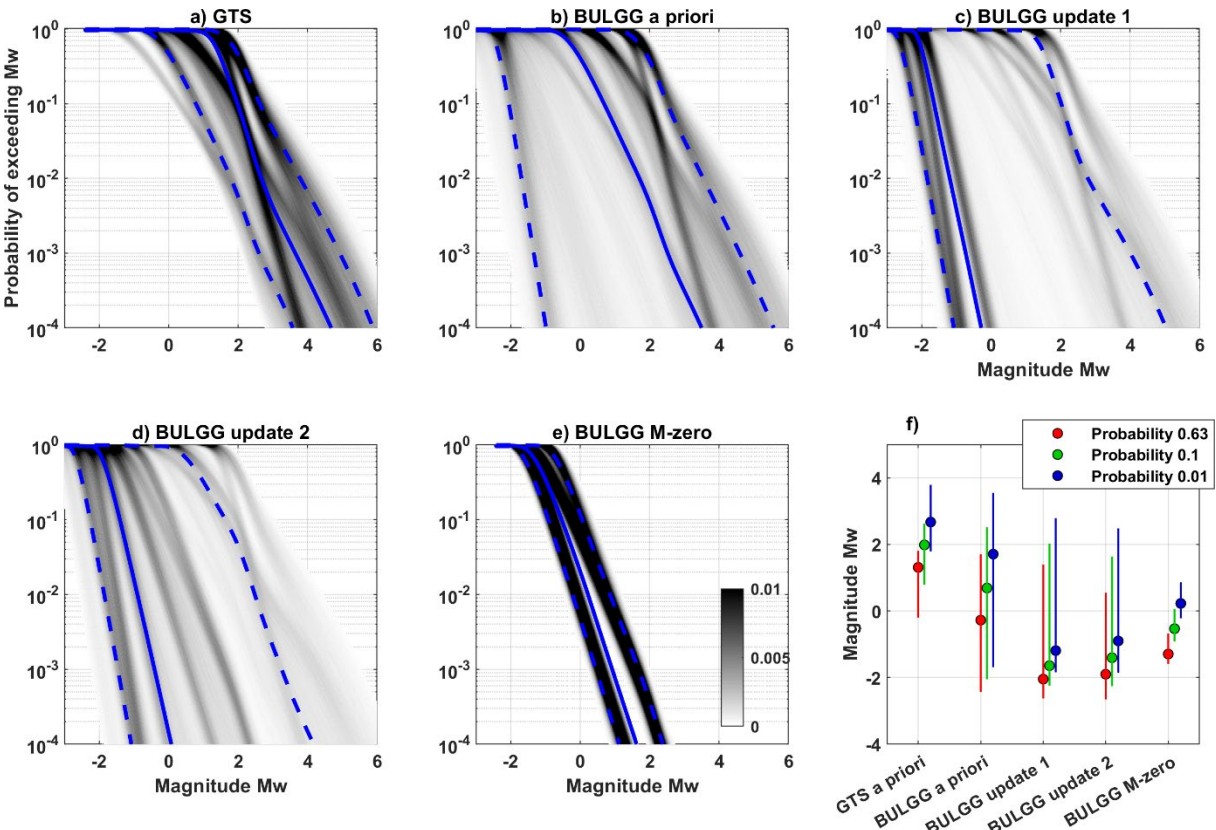

**Figure 11: Hazard curves for all hazard computations expressed in terms of the probability of exceeding magnitude Mw. Injection volume is 100 m³. a) GTS a priori, b) BULGG a priori, c) BULGG update 1, d) BULGG update 2. e) BULGG M-zero. The grey**
**shading represent the PDF, the blue solid line is the median and the blue dashed lines are the 10% and 90% percentile of all solutions. f) Summary of all hazard computations for specific probability levels.**

The outcomes of experiments at GTS and BLUGG in terms of the maximum induced magnitudes are compared against the corresponding predictions of the PISHA (i.e. probability of exceeding a magnitude Mw; Figure 12). Because the GTS a priori analysis relies mostly on deep injections and not on underground laboratory experiments, the maximum observed magnitudes are much below what it predicted (Figure 12a, note that the probability of 0.63% corresponds to a rate of one). Experience

from the GTS experiments (Villiger et al., 2020) now considered in the BULGG a priori analysis still predicts the maximum magnitude induced during the DESTRESS stimulations (Figure 12b). These experiments are considered in the BULGG update 1 with high weights (Figure 6) and lead to much lower magnitude predictions, which are well in agreement with the maximum magnitude observed during the VALTER stimulations (Figure 12c); the maximum observed magnitudes group around the

63% and 10% probability lines for volumes larger than 1 m3. The predictions of the BULGG update 2, now considering the VALTER stimulations are comparable to the BULGG update 2. The maximum magnitude Mw-0.41 during the M-zero





experiment corresponds to a 0.1% probability level (Figure 12d). If only data from stimulation at nearby intervals and at larger volumes (i.e. >5 m$^3$) are considered, the maximum magnitude corresponds to a 10% level (Figure 12e).


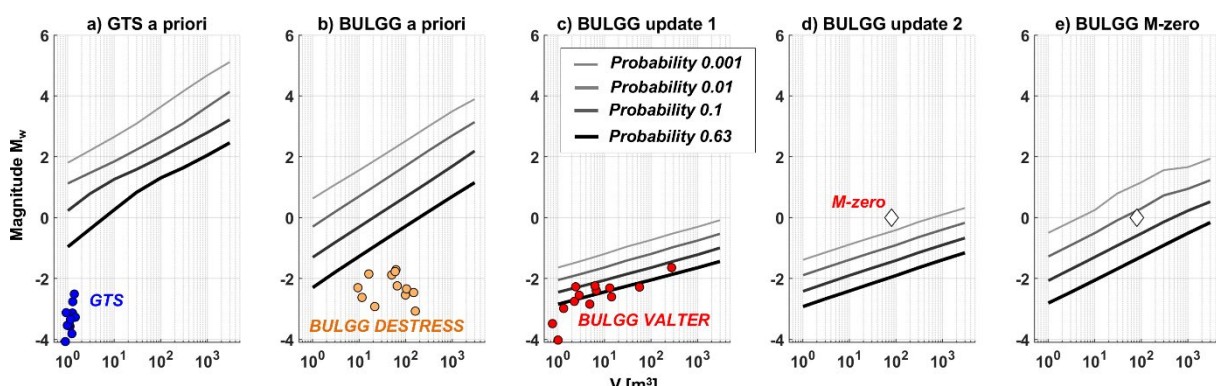

**Figure 12: Magnitude M$_w$, that is exceeded at different probability levels, for different injection volumes. a) GTS  a priori analysis along with maximum magnitudes induced during experiments at the GTS (Villiger et al.,  2020). b) BULGG a priori analysis along with the maximum magnitude induced during DESTRESS stimulations (Giardini et al., 2021; Figure 3c). c) BULGG update 1 along with maximum magnitudes induced during VALTER stimulations (Obermann et al.,  2025, Figure 3d). d) BULGG update 2 and e) BULGG Mzero, both along with maximum magnitude planned to be induced during M-zero experiment.**

*Hazard curves*

The range of possible hazard curves becomes even larger when GMMs are used to compute the probability of exceeding a certain PGV (Figure 13 for injection volume of 100 m$^3$ and a distance from the source of 100 m). The hazard analysis for the GTS, not considering parameters of underground laboratory experiments, results in a range of 2-3 orders of magnitude between the 10% and 90% percentile (i.e. the PGV exceeded at a certain probability level; Figure 13f). If parameter sets of underground laboratories are considered (BULGG a priori analysis), the range becomes unreasonably high and covers up to six orders of magnitude. Both the large range in magnitude probabilities (Figure 11b) and the large uncertainties in the GMMs in the absence of site-specific estimates (Figure 10 a and b) result in an extreme span of hazard estimates. For the BULGG update 1 and 2 analyses locally calibrated GMMs were used that have lower uncertainties (Figure 10c and d). Yet, the range of possible hazard estimates remains high, because the range in magnitude probabilities is already very high. The range of hazard estimates reduces only once locally calibrated GMMs are used along with a$_{fb}$- and b-values of only a few representative hydraulic stimulations, such as used for the BULGG M-zero estimate.

The median strongly depends on the relative weighting of the parameter sets. The median of the expected PGV (i.e. the PGV exceeded with a probability of 0.63) decreases from 3 mm/s to 0.01 mm/s (i.e. factor 300) comparing the GTS a priori to the BULGG updates 1 and 2. For the BULGG M-zero analysis, the median of the expected PGV is 0.04 mm/s.

For the GTS a priori analysis, the probability of exceeding the threshold value of 30 mm/s is about 0.07 (range 4e-4 to 0.76). For BULGG update 2, it ranges from <<1e-7 to 0.03. For the BULGG M-zero, it is 4e-5 (range <<1e-6 to 5e-4)



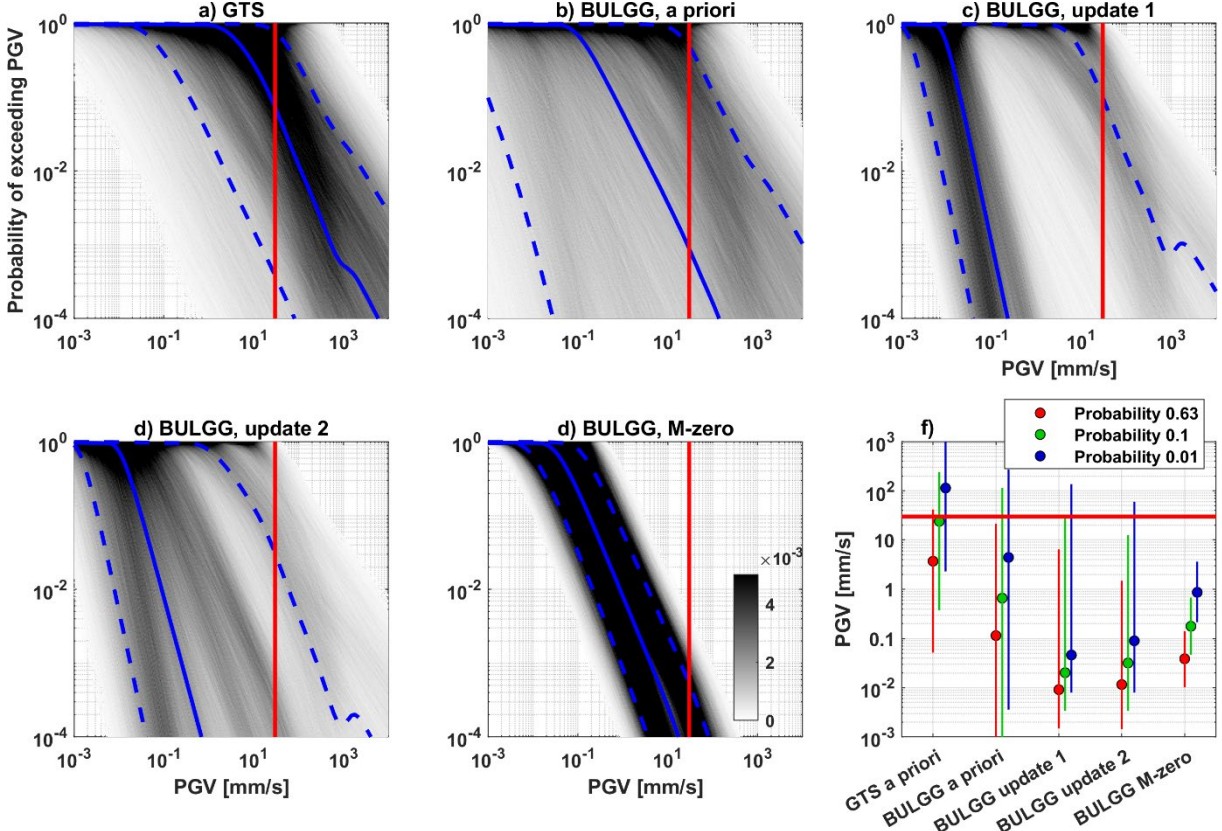

**Figure 13: Hazard curves expressed in terms of probability of exceeding a certain PGV at distance 300 m from the injection. Injection volume is 100 m³. a) GTS a priori, b) BULGG a priori, c) BULGG update 1, d) BULGG update 2. The grey shading represent the PDF, the blue solid line is the median and the blue dashed lines are the 10% and 90% percentile of all solutions. The vertical red lines indicates the PGV threshold 30 mm/s. e) Summary of all hazard computations for specific probability levels. The horizontal red line corresponds to 30 mm/s.**

If hazard is computed for a range of injection volumes and distances (e.g. Cremen and Werner, 2020), the results may be used for designing stimulation experiments based on a map of PGV values that are exceeded at a predefined probability level as a function of distance and volume (Figure 14). For instance, if a potentially damaging PGV of 30 mm/s must not be exceeded with a probability of 0.01, we find that the GTS a priori analysis indicates that injecting 1 m³ at 10 – 30 m distance may already exceed this level. Refining the analysis using underground laboratory experience, the injection volume may be much larger. In the case of the BULGG update 2, the threshold is not even reached for volume up to 3000 m³ regardless of distance. Thus, also for the critical Furka tunnel railway infrastructure, seismic hazard is very small for volumes up to 3000 m³. Again, the situation is different if the 90%-percentile instead of the median of all solutions is considered; even for the BULGG update 2, the threshold is already exceeded with 30 m³ injected at 100 m distance. The difference between the median and 90% percentile maps is smaller for the BULGG M-zero analysis. At 100 m distance, 30 mm/s is not exceeded for volumes below 1000 m3 (90% percentile) or larger volume (median).





A similar strategy is presented in a map of the probability of exceeding PGV=30 mm/s as a function of injection volume and distance Figure 14. In this map, the probability contours of 0.63, 0.1, 0.01 and 0.001 are shown.



**Figure 14: Map of induced seismic hazard estimates as a function of distance and injection volume. Distance range from 10 m to 10'000 m and volume from 1 to 3000 m³. a) GTS a priori hazard computation, b) the BULGG a priori, c) the BULGG update 1 and d) the BULGG update 2 hazard computation. In the columns of figures the PGV exceeded with a probability of 0.01 based on the median and the 90% percentile of the hazard estimate, as well the probability of exceeding 30 mm/s again based on the median and the 90% percentile are shown.**



## 7.   Discussion

### *Sensitivities and uncertainties*

Our seismic hazard computation for hydraulic stimulations in BULGG highlights the benefits of PISHA, which lies in the quantitative and transparent representation of the uncertainties by considering the experience of a wide range of induced
seismicity cases. In this way, our PISHA also sheds light on the deficiency in our capability to predict induced seismic hazard, not only specifically for BULGG but also in general, i.e. also for full-scale geothermal projects. The wide spread of possible hazard estimates in Figure 11 and Figure 13 illustrates that drawing from parameter sets of world-wide induced seismicity datasets and from GMMs stemming from various contexts (mining, induced seismicity, etc.) does not produce converging hazard estimates, but rather adds to enlarge uncertainties.

The GMMs used before BULGG-specific observations on ground motions become available (Figure 10a and b) predict possible PGV for a given magnitude that exceeds two orders of magnitude, the cause lying both in the epistemic uncertainty (here represented by using several possible GMMs) and in the aleatory uncertainty (represented by considering the inherent uncertainty of each GMM). These GMMs are all based on seismicity data with a magnitude range of $M_w>1.0$. Although this is the range that is most relevant for hazard in the underground laboratory context (i.e. relevant ground motions occur around
these magnitudes, Figure 10), the extrapolation to lower magnitudes brings additional uncertainties into the hazard computation (e.g. in Figure 13) as it covers the entire range of magnitudes and possible GMMs. Clearly, epistemic uncertainty is reduced by choosing GMMs based on local observations (Figure 9, Figure 10c and d); the massive uncertainty in the ground motion hazard curves of the BULGG a priori analysis (Figure 13b) is somewhat reduced once the uncertainty in the GMMs reduces for the BULGG update 1 and 2 analyses (Figure 13c and d).

GMMs retrieved from local data would further reduce uncertainty to some degree. The difficulty lies in covering a large range in terms of magnitude and distance, which may be addressed by combining induced seismicity data with data from active seismic experiments. However, there is also an aleatory component to the uncertainties that is inherent to any ground motion estimate and is related to source complexity (radiation pattern, stress drop, etc), to path effects and to effects close to the recording instrument. For instance, the uncertainty (i.e. the standard deviation) in the equation by Douglas et al., (2013)
produces a factor of 7, which would amount to more than two orders of magnitude uncertainty in the PGV estimate (10% to 90% percentile). It is noteworthy that this already represents epistemic uncertainty, too, because Douglas et al., (2013) derive the GMM from a combination of six induced seismicity case studies. The other GMMs used here predict uncertainties represented by factors of two to three (standard deviation). Thus, it is important to bear in mind that local GMMs will reduce the total uncertainties by reducing the epistemic uncertainties, but aleatory uncertainties producing an order of magnitude
variability in the predicted PGV will remain.

However, as the sequence of the PISHA studies in Figure 11 shows, the primary uncertainty in the hazard computation lies in the range of seismogenic response represented primarily by the $a_{fb}$- and b-value. While the $a_{fb}$- and b-value of deep injections from typically larger volume injections cover a relatively small range ($a_{fb}$: -3.2 to 0.19, b: 0.65 – 1.58), the underground



laboratory experiments cover a much larger range ($a_{fb}$: -10.5 to -1.6, b: 0.95 to 2.95). The challenge to integrate such variability

in a probabilistic analysis comes down to assigning weights to the individual parameter sets, which must rely on expert judgment (Figure 6). Adhering to the principles of PISHA, site-specific information may not replace but down-weight non-site-specific information. In the BULGG update 1 and 2, BULGG parameters receive 75% weight, which makes them dominate the median hazard estimate. However, the weights have a strong impact on the median of the hazard estimates (Figure 11f), but a much lesser impact on the 90% percentile of the estimates. The enormous uncertainties in magnitude probabilities, which

are even more pronounced for ground motion probabilities (Figure 13f) only reduce once a limited range of seismogenic responses are considered (i.e. BULGG M-zero analysis).

  In conclusion, we can state that uncertain $a_{fb}$- and b-values have by far the greatest impact on the uncertainties of the hazard computations. Uncertain GMMs are second in explaining the overall uncertainties. $M_{max}$ has a rather small impact in comparison as has been already observed by Mignan et al., (2015).


### Scale- and depth-dependent seismogenic response

  The ranges of $a_{fb}$- and b-value in Figure 7c and d raise the question, whether hazard computation across many orders of magnitude of injection volume is feasible, specifically if hazard estimates for underground laboratories from deep injections or vice-versa are possible. Despite the limited amount of data in Figure 7, there is a tendency for deep injections towards higher

$a_{fb}$- and lower b-values, although the smaller-volume injections in underground laboratories overlap with this range, but also contain low $a_{fb}$- and high b-values. Upscaling of underground laboratory experience may be limited, because the smaller-volume injections access a more limited rock volume and sense a seismogenic response that is determined by local fracture network properties. In contrast, injecting larger volumes produces seismicity that is dominated by a stronger seismogenic response of the most critically stressed and most conductive fractures in the stimulated volume, which may otherwise be missed

if smaller volumes are injected, and a more limited fracture network is accessed.

  Additionally, lower stress levels at shallower depths of underground laboratories may also lead to more benign seismogenic responses. For tectonic earthquakes, a primary cause for variable b-values is seen in the stress field. Typically, seismicity in the uppermost 3-4 kilometers exhibits higher b-values (e.g., Spada et al. (2013) than below and possibly lower a-values as seismicity decreases towards shallow depths. Petruccelli et al. (2019) find that b-values depend both on depths and stress

regime. Scholz (2015), interpreting the depth dependence by Spada et al. (2013), suggested that b-values depend on differential stress. A linear extrapolation of Scholz's relationship towards small differential stresses (on the order of 5 MPa and 10 MPa at the GTS and BULGG respectively; Krietsch et al. (2018); Bröker and Ma, (2022) would imply a b-value of 1.2 or higher. Apart from the stress conditions, the frictional properties of faults influenced by their material and structural properties as well as their genesis (e.g. McClure and Horne, 2013) may additionally define the seismogenic response. Note, however, not only

variable b-values, but also a break-down of the Gutenberg-Richter distribution assuming constant b-values has been observed for many cases (e.g. Villiger et al., 2020, 2021; Urban et al., 2016; GSK, 2018), which may be of mechanical origin. The





importance of controlled underground experiments to shed light on these dependencies is highlighted by the fact that most studies on a- and b-values stem from tectonic earthquakes (i.e. greater depths and stress levels). The reason for more scatter and weaker seismogenic response in underground laboratory experiments deserves more investigation, because understanding

the underlying reasons may open doors to safer stimulations in the deep underground.

However, the causes for the high b-value and low $a_{fb}$-values in Figure 11 may not only be physical but could also be instrumental and thus apparent. Various studies on seismicity at magnitude levels much below Mw0.0 stress the difficulty of estimating reliable earthquake magnitudes (Kwiatek et al., 2011; Manthei and Plenkers, 2022). The issue is also illustrated by the deviation between the moment magnitude Mw and local magnitude $M_L$ observed for magnitudes $M_w$<2.0-3.0. The deviation

is associated with the relationship between source properties and attenuation properties, and is held responsible for changing b-values at different magnitude levels (Deichmann, 2017). Similarly, Wesseloo (2016) points out that the shape of the frequency-magnitude-distribution (FMD) may be affected by the sensor bandwidth, with the FMD (i.e. apparent b-value) becoming steeper if the sensor eigenfrequency cuts the spectrum above the corner frequencies at higher magnitudes. Thus, predicting seismicity with Mw>0.0 from earthquake magnitude distributions of much smaller events requires that source

characterization of these small earthquakes is carefully considered and take into account seismic attenuation at the relevant levels and the instrumental responses of sensors typically used at these levels. Again, underground laboratory experiments are the opportunity to overcome instrumental challenges and to eventually bridge the seismological scales (Gischig et al., 2020).

### *Traffic light system*

While the median hazard estimates represent the hazard level based on the wide range of possibilities, it is common practice to design engineering endeavours using the 90% or 95% percentile (or even the worst-case) instead of the median (e.g. Cai and Kaiser, 2018). In our case, the BULGG update 2 gives clearance to any injection volumes (Figure 14), but the 90% percentile still indicates a chance of inducing a damaging event at a distance of 100 m. Although the median of the BULGG M-zero analysis is higher than for the BULGG update 2, the hazard represented by the 90% percentile is lower and indicates

that the chances for damaging events are very low even with high injection volumes of 3000 m$^3$.

Despite the discrepancy between the median and 90% percentile of the hazard curves, originating from the substantial uncertainties discussed earlier, it is recommended to incorporate them both in hazard-relevant decisions. By doing so, we acknowledge that induced seismicity can hold surprises (e.g. as the cases of Pohang and St. Gallen have shown) and that these have to be anticipated regardless of how thoroughly the hazard is estimated. The uncertainty in hazard estimates also highlights

the importance of updating induced seismic hazard analysis as soon as site-specific information becomes available, as proposed by Wiemer et al. (2018). This means not only between project phases (as done here) but preferably even in near-real time if a corresponding workflow in the framework of an adaptive traffic light system (ATLS) is in place (e.g. Kiraly et al., 2018; Broccardo et al, 2019; Zhou al., 2024; Ritz et al., 2024). In the presented case, the BULGG updates 1 and 2 resulted in lower





hazard levels as anticipated in the a priori studies thus giving way to presumably safe experimental work. However, updating

is even more important if the observed seismic response starts indicating higher levels of seismicity.

For hydraulic stimulation in the BULGG, a TLS with multiple layers is proposed (Figure 15). The first layer consists of fixed thresholds in terms of PGV: green/yellow: 0.5 mm/s, yellow/orange: 2.5 mm/s, orange/red 15 mm/s. Note that the PGV of 15 mm/s still leaves a safety margin to the damaging threshold of 30 mm/s. Using the GMMs in Figure 10c and d, these translate into magnitude thresholds that depend on the distance of the hydraulic stimulation to the experiment cavern, which is the

second TLS layer. At 100 m the corresponding thresholds are Mw0.0, 0.8 and 1.7, at 300 m Mw0.8, 1.6 and 2.5 (Figure 15a and b). These thresholds correspond to traditional TLS reported in the literature (see Introduction). An alternative third layer (Figure 15c and d) includes probabilistic thresholds similar to those proposed by (Mignan et al.,2017). The concept relies on defining a threshold magnitude Mw(safe) that must not be exceeded. Here, we used the magnitudes at the orange/red threshold: Mw(safe) = 1.7 at 100 m and Mw(safe) = 2.5 at 300 m distance. Using a target injection volume of, for instance, 1000 m$^3$,

one can produce a map of the probability of exceeding Mw(safe) as a function of the $a_{fb}$- and b-values. The different traffic light colors correspond to the probability levels 0.001, 0.01 and 0.1. The probability map serves as the basis for an adaptive TLS, in which hazard can be evaluated as soon as the $a_{fb}$- and b-value of the induced seismic sequence are determined. This can be done during stimulations, provided that reliable magnitudes can be estimated (Mesimeri et al., 2024), or after different phases of the stimulation, for instance after a test stimulation with only a fraction of the target injection.






**Figure 15: a, b) Fixed TLS thresholds shown with the GMMs at 100 m and 300 m in the background. The blue lines are the median (solid) and the 10% and 90% percentiles (dashed) b) Probabilistic TLS levels for 1000 m³ at a distance of 100 m and 300 m. Colors indicate the probability of exceeding a predefined magnitude Mw(safe).**

## 8. Conclusions

We here propose a workflow for a probabilistic analysis of induced seismic hazard during hydraulic stimulations, which can be quickly updated as soon as new information becomes available. Resulting hazard estimates are presented in a series of diagnostic visualizations that support the design of hydraulic stimulations and the mitigation strategies for induced seismic risk. For the ongoing stimulation experiments at the BULGG, our hazard computations show that injections of 100 m³ at distances of 100 to 300 m from the experimental cavern are acceptable with a probability of exceeding a PGV of 30 mm/s being P(PGV>30mm/s)<0.001. The sequence of hazard computations, which include more site-specific information on the seismogenic response at the BULGG in a step-wise manner, also highlights the sensitivies of the hazard computation on the seismogenic response parameters and ground motion prediction equations. The range of possible seismogenic responses



(expressed by $a_{fb}$- and b-values) documented for world-wide case studies seem to cover a different range as underground laboratory experiments at shallower depths. Together with uncertainties in GMMs, if they are not calibrated at the site, produce an enormous spread of possible hazard estimates. This illustrates the importance of collecting site-specific data on both the seismogenic response and GMMs. Additionally, the weighting of different parameter sets regarding their relevance to our

specific BULGG experiments results in additional uncertainty in the hazard estimates further, highlighting that a more profound seismo-hydromechanical understanding is required for assessing induced seismic hazard a priori. Currently, uncertainties can only be addressed by reevaluating the hazard at different project stages, and by accompanying the stimulations itself with a hazard mitigation scheme (e.g. a traffic lights system) that allows to anticipate and appropriately react upon induced seismic surprises. Ideally, the scheme adapts the concept of an ATLS that allows the processing of incoming

new seismicity data as a basis of hazard computation in near-real-time.

The stimulation experiments in underground laboratories (GTS, BULGG, Aspö, etc) indicate that the seismogenic response at depths of 500 – 1000 m may be substantially weaker compared to injections at depths of several kilometers. While this may question the transferability of underground laboratory research to full-scale operations, and it also holds promise that if we understand the underlying cause of the weaker seismogenic response, it may light the way to safer exploitation of geoenergy

resources. In any case, underground laboratory experiments are a safe way to perform reservoir geomechanics research from a seismic hazard perspective.

## 9.  Data availability

Data from seismic sequences used in this study are given in the Appendix. Seismicity catalogues of the VALTER project have been published by Rosskopf et al (2024b). Seismicity catalogues can be retrieved from the main authors and will be published

on a dedicated repository upon acceptance of the paper.

## 10.  Author contributions

VSG, ARP, SW conceptualized the study. VSG and ARP performed the analysis and wrote the paper. MB, FC and AM contributed to the analysis, the parameters of case studies and to the methodology. AA, FB, PM, RC, DK and FS performed the DESTRESS and ZoDrEx experiments and contributed the seismic catalogues and hydraulic data. MH and RB enabled

experimental work in the BULGG. KB, XM, VCR, NGD, VD, PK, KP, LV, QW, AZ, AO, MM, MR and AS contributed to instrumentation of the experiments, conceptualized and performed the hydraulic stimulations, analyzed hydraulic data and prepared seismic catalogues. PAS MAM, AO, MM, KB, LV, and VSG contributed to planning of the FEAR experiments. JA, HM, DG supervised the projects.



## 11. Competing interests

The authors declare that they have no conflict of interest.

## 12. Acknowledgments

In the "Bedretto Underground Laboratory for Geosciences and Geoenergies", ETH Zurich studies in close collaboration with national and international partners techniques and procedures for a safe, efficient, and sustainable use of geothermal heat and 695 questions related to earthquake physics. The BedrettoLab is financed by the Werner Siemens Foundation, ETH Zürich and the Swiss National Science Foundation (SNSF). The research in this publication was conducted within the P&D BFE project VALTER (SI/501496-01), the EU project DESTRESS (691728), the Swiss part of the GEOTHERMICA project ZoDrEx (SI/501720), the SNSF Multi PhD project 200021_192151, as well as the ERC Synergy FEAR 856559. The BedrettoLab would like to thank Matterhorn Gotthard Bahn for providing access to the tunnel.





## 13. Appendix A

**Table A1: Collection of model parameters for a range of different case studies. From [1]Mignan et al.,2017), re-estimated for this study; [2]Dinske and Shapiro (2013), [3]Kiraly et al., (2014), re-estimated for this study; [4]Albaric et al., (2014); [5]Villiger et al., (2020); [6]Kwiatek et al., (2018); [7]Broccardo et al., (2020) all others estimated for this study.**

| Case study | Stimulation | $m_c$ | b | $a_{fb}$ | % of events after shut-in |
|---|---|---|---|---|---|
| 1 | [3]St. Gallen, 2013 | 0.2 | 1.08 | -0.07 | |
| 2 | [1]Basel, 2006 | 0.8 | 1.58 | 0.19 | 31 |
| 3 | [1]Garvin, 2011 | 1 | 0.77 | -1.52 | 14 |
| 4 | [1]KTB, 1994a | -1.5 | 0.98 | -1.41 | 24 |
| | [1]KTB, 1994b | -1.4 | 0.87 | -1.56 | 27 |
| | [1]KTB, 2000 | -0.8 | 1 | -2.25 | 7 |
| 5 | [1]Paradox Valley, 1994 | 0.6 | 1.08 | -2.42 | 3 |
| | [1]Paradox Valley, 2008 | 0.4 | 0.76 | -2.77 | 1 |
| 6 | [1]Newberry, 2012 | 0.2 | 0.8 | -1.56 | 57 |
| | [1]Newberry, 2014a | 0 | 0.98 | -1.02 | 10 |
| | [1]Newberry, 2014b | 0.2 | 1.05 | -1.58 | 16 |
| 7 | [1]Soultz, 1993a | -1.4 | 0.89 | -1.83 | 5 |
| | [1]Soultz, 1993b | -1.1 | 0.99 | -2.24 | 29 |
| | [1]Soultz, 2000 | 0.1 | 0.98 | -0.3 | 19 |
| | [1]Soultz, 2004 | -0.3 | 0.83 | -0.61 | 15 |
| 8 | [7]Cooper Basin, 2003 | -0.7 | 0.79 | -0.9 | 6 |
| 9 | [4]Paralana, 2011 | -0.3 | 1.32±0.02 | 0.1 | |
| 10 | [2]Ogachi, 1991 | | 0.74 | -2.65±0.1 | |
| | [2]Ogachi, 1993 | | 0.81 | -3.2±0.3 | |
| 11 | Pohang 2017 | 0.7 | 0.65 | -2 | |
| 12 | [5]Grimsel HS2, 2017 | -4.32 | 1.69±0.26 | -5.8 | 6.8 |
| | [5]Grimsel HS4, 2017 | -4.32 | 1.36±0.04 | -3.0 | 2.3 |
| | [5]Grimsel HS5, 2017 | -4.32 | 1.03±0.05 | -2.4 | 4.6 |
| | [5]Grimsel HS3, 2017 | -4.32 | 1.93±0.37 | -7.6 | 17.8 |
| | [5]Grimsel HS8, 2017 | -4.32 | 1.61±0.12 | -4.9 | 8.7 |
| | [5]Grimsel HS1, 2017 | -4.32 | 1.93±0.39 | -6.6 | 7.7 |
| | [5]Grimsel HF3, 2017 | -4.32 | 1.55±0.26 | -4.8 | 2.9 |
| | [5]Grimsel HF2, 2017 | -4.32 | 1.35±0.08 | -4.0 | 7.6 |
| | [5]Grimsel HF8, 2017 | -4.32 | 2.66±0.36 | -9.0 | 3.9 |
| 13 | [6]Aspö, 2017 | -4.1 | 2.9±0.2 | -8.65 | 25 |
| 14 | BULGG, CB1 | see values in Table A2 | | | |
| | BULGG, DESTRESS ST1 | see values in Table A2 | | | |
| | BULGG, DESTRESS ST2 | see values in Table A2 | | | |
| 15 | BULGG, VALTER ST1 | see values in Table A2 | | | |






**Table A2: Summary of seismicity characteristics of all hydraulic stimulations in BULGG. *Volume refers to the volume that was injected into the fracture network, which is less than the total injected volume in case a **bypass along the packers has been identified. Details of the different stimulations procedures and projects can be found in the final report of VALTER[1] (Giardini et al., 2022), ZoDrEx[2] (Meier and Christe, 2023) and in Obermann et al (2024)[3].**

| Borehole | Interval | Stage | Date | Depth [m] | Volume [m³]* | Bypass** | Mc | b | $a_{fb}$ | τ | Located events | #Events > Mc | #After shut-in > Mc | $M_{max}$ |
|---|---|---|---|---|---|---|---|---|---|---|---|---|---|---|
| *Test hydraulic stimulation with packers in borehole CB1 (GES)[1]* | | | | | | | | | | | | | | |
| CB1 | 6 | 1 | 05.02.2020 | 288.5 - 298.5 | 4.86 | - | -3.83 | 2.28 (1.80–2.73) | -7.87 | 1301 | 69 | 41 | 3 | -2.99 |
| CB1 | 7 | 1 | 06.02.2020 | 264.0 - 274.0 | 4.47 | - | -3.95 | 2.48 (2.20–2.78) | -8.25 | 401 | 266 | 177 | 17 | -3.29 |
| **CB1** | **ALL** | **1** | **05.02.2020-06.02.2020** | **264.0 - 298.5** | **9.33** | **-** | **-3.93** | **2.55 (2.30 – 2.80)** | **-8.67** | **-** | **335** | **227** | **-** | |
| *Hydraulic stimulations for DESTRESS with packers in borehole ST2 (GES)[1]* | | | | | | | | | | | | | | |
| ST2 | 1a | 1+2 | 11.11.2020-13.11.2020 | 306-312 | 49.69 | 56.00% | -3.00 | 2.48 (2.18 – 2.78) | -6.90 | - | 287 | 166 | - | -1.88 |
| ST2 | 1b | 1 | 30.11.2020 | 304.8-312 | 12.28 | 56.50% | - | - | - | - | 7 | - | - | -2.78 |
| ST2 | 2a | 1 | 17.22.2020-19.11.2020 | 313.6-319.6 | 11.43 | 44.00% | - | - | - | - | 19 | - | - | -2.62 |
| ST2 | 2b | 1 | 29.11.2020 | 312.16-319.36 | 4.70 | 65.50% | - | - | - | - | 4 | - | - | -2.89 |
| ST2 | 4a | 1+2 | 21.11.2020-22.11.2020 | 319.2-327.6 | 16.01 | - | -3.11 | 2.48 (2.18 – 2.85) | -6.82 | - | 218 | 126 | - | -1.85 |
| ST2 | 4b | 1 | 30.11.2020 | 319.4-326.4 | 12.15 | - | -2.93 | 2.48 (2.10 – 2.85) | -6.37 | 1301 | 180 | 98 | 3 | -1.79 |
| ST2 | 5 | 1+2 | 23.11.2020-25.11.2020 | 325.22-333.72 | 61.26 | 11.00% | -2.95 | 2.10 (1.90 – 2.30) | -5.52 | - | 511 | 297 | - | -1.71 |
| ST2 | 6 | 1+2 | 27.11.2020-29.11.2020 | 335.2-345 | 58.99 | - | -3.09 | 2.05 (1.78 – 2.38) | -6.00 | - | 208 | 127 | - | -1.77 |
| **ST2** | **ALL** | **-** | **11.11.2020-30.11.2020** | **306-345** | **226.48** | **variable** | **-3.01** | **2.23 (2.10 – 2.35)** | **-6.12** | **-** | **4509** | **861** | **-** | **-1.71** |
| *Hydraulic stimulations for DESTRESS with packers in borehole ST1 (GES)[1]* | | | | | | | | | | | | | | |
| ST1 | 10 | 1 | 13.12.2020 | 268.74-277.68 | 21.51 | - | - | - | - | - | 4 | - | - | -2.92 |
| ST1 | 11 | 1 | 12.12.2020-13.12.2020 | 278.67-287.61 | 98.08 | - | -2.7 | 2.7 (2.05-3.30) | -7.97 | - | 60 | 20 | 0 | -2.54 |
| ST1 | 12 | 1+2 | 17.12.2020-18.12.2020 | 288.00-301.00 | 65.15 | variable | -2.76 | 3.35 (2.98 – 3.70) | -8.87 | - | 301 | 152 | - | -2.24 |
| ST1 | 13 | 1 | 16.12.2020-17.12.2020 | 298.54-307.48 | 103.34 | - | -2.66 | 3.38 (2.90 – 3.73) | -9.05 | - | 254 | 87 | 0 | -2.34 |
| ST1 | 14b | 1 | 16.12.2020 | 311.00-321.00 | 6.18 | - | - | - | - | - | - | - | - | - |
| ST1 | 15 | 1 | 14.12.2020-15.12.2020 | 321.88-330.82 | 159.74 | 34.50% | - | - | - | - | 1 | - | - | -3.07 |





| Borehole | Interval | Phase | Date | Depth range | Volume | % | | Mag. range | | Inj. | | | | |
|---|---|---|---|---|---|---|---|---|---|---|---|---|---|---|
| ST1 | 16a | 1 | 14.12.2020 | 335.28-344.22 | 7.23 | 49.00% | - | - | - | - | - | - | - | - |
| ST1 | 16b | 1 | 18.12.2020-19.12.2020 | 335.28-344.24 | 139.61 | - | -2.86 | 3.23 (2.83 – 3.58) | -9.3 | - | 197 | 117 | 0 | -2.46 |
| **ST1** | **ALL** | - | **12.12.2020-19.12.2020** | **268.74-344.24** | **599.43** | **variable** | **-2.82** | **2.95 (2.78 – 3.20)** | **-8.4** | - | **3074** | **500** | - | **-2.24** |
| *Hydraulic stimulations for VALTER with sliding sleeves ST1, Phase 0 (GES)[2]* | | | | | | | | | | | | | | |
| ST1 | 1+2 | 1+2+3 | 02.05.2021 | 366.13-385.53 | 52.994 | - | - | - | - | - | 15 | - | - | -1.7 |
| ST1 | 4 | 1+2 | 04.05.2021 | 336.45-344.87 | 63.632 | - | - | - | - | - | 7 | - | - | -2.5 |
| ST1 | 6 | 1+2+3 | 05.05.2021 | 254.67-307.31 | 57.504 | - | - | - | - | - | 62 | - | - | -1.5 |
| *Hydraulic stimulations for ZoDrEx with packers/notch in borehole ST2 (GES)[2]* | | | | | | | | | | | | | | |
| ST2 | 6 | 1 | 21.05.2021-23.05.2021 | 332.52-350.90 | 53.2 | - | - | - | - | - | 80 | - | - | -2.1 |
| ST2 | 1 | 1+2 | 06.10.2021-07.10.2021 | 306.5 | 13.573 | - | - | - | - | - | 6 | - | - | -2.2 |
| ST2 | 8 | 1+2+3 | 08.10.2021-11.10.2021 | 283.75 | 21.015 | - | - | - | - | - | 43 | - | - | -2.52 |
| ST2 | 7 | 1 | 08.10.2021-11.10.2021 | 276 | 0.021 | 95% | - | - | - | - | - | - | - | - |
| ST2 | 4 | 1 | 08.10.2021-11.10.2021 | 324.6 | 5.103 | 90% | - | - | - | - | - | - | - | - |
| *Hydraulic stimulations for VALTER with sliding sleeves ST1 Phase 1 stimulations (ETH)[3]* | | | | | | | | | | | | | | |
| ST1 | 7 | 1+2 | 17.11.2021 | 218.26-253.32 | 14.1 | - | -4.2 | 1.15 (1.00 - 1.28) | -3.85 | 1700 | 262 | 179 | 22 | -2.6 |
| ST1 | 8 | 1+2 | 09.02.2022 | 186.68-216.76 | 4.8 | - | -4.04 | 2.45 (2.25 – 2.60) | -7.88 | 200 | 1309 | 563 | 32 | -2.84 |
| ST1 | 9 | 1+2 | 16.02.2022 | 170.82-185.15 | 1.32 | - | -4.48 | 2.35 (2.13 – 2.60) | -8.33 | 500 | 572 | 243 | 17 | -2.98 |
| ST1 | 10 | 1+2 | 02.03.2022 | 151.98-169.32 | 0.76 | - | -4.58 | 2.23 (2.08 – 2.43) | -7.48 | 300 | 622 | 434 | 17 | -3.48 |
| ST1 | 11 | 1+2 | 09.03.2022 | 132.18-150.47 | 2.24 | - | -4.25 | 1.53 (1.25 – 1.88) | -5.08 | 600 | 98 | 63 | 4 | -2.75 |
| ST1 | 12 | 1+2 | 16.03.2022 | 123.18-130.68 | 0.36 | - | -4.42 | 0.95 (0.83 – 1.08) | -1.55 | - | 233 | 164 | 1 | -2.42 |
| ST1 | 13 | 1+2 | 23.03.2022 | 103.43-121.67 | 12.87 | - | -4.11 | 1.20 (1.13 – 1.25) | -2.98 | 300 | 2444 | 1295 | 85 | -2.31 |
| ST1 | 14 | 1+2 | 30.03.2022 | 47.17-101.93 | 1 | - | -4.42 | 2.83 (2.45 – 3.00) | 10.55 | - | 204 | 87 | 0 | -4.02 |
| **ST1** | **ALL** | | **17.11.21-30.03.22** | **47.17-253.32** | **37.26** | - | **-4.21** | **1.35 (1.28 – 1.38)** | **-3.78** | | **5744** | **3054** | - | **-2.31** |
| *Hydraulic stimulations for VALTER with sliding sleeves ST1 Phase 2 stimulations (ETH)[3]* | | | | | | | | | | | | | | |
| ST1 | 8 | 1+2 | 22-23.06.2022 | 186.68-216.76 | 274.15 | - | -4.14 | 1.10 (1-05 – 1.13) | -3.28 | 7500 | 9498 | 5678 | 201 | -1.64 |





| | | | | | | | | | | | | | | |
|---|---|---|---|---|---|---|---|---|---|---|---|---|---|---|
| ST1 | 9+10 | 1 | 14.03.2023 | 151.98-185.15 | 56.17 | | -4.19 | 1.43 (1.38 – 1.45) | -4.20 | 3000 | 6063 | 3867 | 233 | -2.28 |
| ST1 | 11 | 1+2 | 18-19.04.2023 | 132.18-150.47 | 6.61 | | -4.29 | 1.55 (1.50 – 1.60) | -4.18 | 1200 | 3853 | 2174 | 62 | -2.39 |
| ST1 | 12 | 1+2 | 06-07.07.2022 | 123.18-130.68 | 2.39 | | -4.37 | 1.33 (1.20 – 1.48) | -3.80 | - | 420 | 236 | 0 | -2.27 |
| ST1 | 11 | 1+2+3 | 12.07.2023 | 132.18-150.47 | 6.22 | | -4.14 | 1.33 (1.28 – 1.35) | -2.93 | 1100 | 4643 | 2741 | 243 | -2.24 |
| ST1 | 12 | 1+2 | 28.02-02.03.2023 | 123.18-130.68 | 2.84 | | -4.35 | 1.68 (1.50 – 1.80) | -5.30 | - | 605 | 299 | 2 | -2.55 |



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
