# Peer review of "Updating induced seismic hazard assessments during hydraulic stimulation experiments in underground laboratories: workflow and limitations"

_EGUsphere, 2024_

## Author Comment (AC1)

**Reponses to the reviewer's comment 1 on manuscript egusphere-2024-3882**

*Review by Martin P. Mai, 15 Feb. 2025*

*Dear Martin Mai,*

*We'd like to thank you for your thorough review and the supportive comments. We agree that the paper has to cover a wide range of topics to explain the workflow and results of our probabilistic induced seismicity analysis. You comments helped to clarify many aspects of our analysis.*

**General Comments**

Induced seismicity in context of enhanced oil & gas exploitation, wastewater injection, and geothermal-energy harvesting is a recurring problem that operators, regulators and nearby communities have to deal with. In case of geothermal energy, hydraulic simulations have led to induced seismicity at a level that caused substantial shaking such that operations were stopped because the associated seismic hazard was not tolerable any more. The question then arises if this (time-dependent) seismic hazard can be quantified and "controlled" during hydraulic stimulation.

The study by Gischig and colleagues examines this question, with a focus on hectometer-scale stimulations in the Bedretto Underground Laboratory for Geoenergies and Geoscience (BULGG) in Switzerland. Inspired from observations and lessons learned in several geothermal projects around the globe, the authors develop a work-flow to compute/update the seismic hazard at a location of interest given fluid-injection parameters and the overall boundary and initial conditions at the site in terms of geology, seismotectonics & regional stresses, whereby the seismic-hazard updates are based on the known injection history and measured seismicity parameters. Noting that hazard estimates may vary greatly depending on the state of information/data and can be better constrained with more data and refined seismicity parameters, the authors also stress that site-specific ground-motion data and a related ground-motion model (GMM) are critical to narrow down the hazard estimates to plausible ranges. The study concludes with proposing an adaptive traffic light system (aTLS) that capture the time-dependent seismic hazard changes in near-real time.

The manuscript is well written, with very accessible graphics and a well-composed structure that naturally navigates the reader through the rather comprehensive material in terms of previous studies, the site of interest, related observations, models developed in the past and applied for the chosen case study, hazard calculations and how these are eventually embedded into a traffic light system. That is, the paper is rich. It is dense. It contains a lot of information that the reader needs to digest. In my view, the authors did an excellent job in this regard, but, ***I do remark that for most 1st to 2nd year graduate students in this field and also the "general but interested reader", this paper may not be an easy read***.

From a technical point of view, I don't have any major comments and concerns. The science is solid. The methods are well known (but not all explained in detail, hence readers need sufficient background knowledge), the data are exquisite, and the overall goal of the study is of importance scientifically and from a socio-economic point of view. Nevertheless, I have several remarks and questions related to the presentation, **level of detail provided on certain aspects of the study (sometimes too much, and thus distracting from the "big picture"'; sometimes too sparse to be able to follow)**, and a few editorial remarks.

The one major point I would like to raise is the use of peak ground velocity (PGV) instead of peak ground acceleration (PGA) as ground-motion intensity measure. The earthquakes consider here are predominantly of small magnitudes and clearly dominated by high-frequency seismic radiation. The authors also state the most earthquakes studied radiate above 10 Hz. ***A rule-of-thumb is that PGV***

*captures shaking intensities for waves around 1 Hz. Hence, the use of PGV is counter-intuitive and perhaps not physically justifiable. This aspects needs detailed consideration and explanation (see below for more on this).*

Overall, I rate this paper as "publishable after minor revisions". New calculations/analyses or substantial re-organization/rewriting are not needed, but I ask for clarifications and editorial improvements that should be straightforward to implement.

In summary, this is a very interesting and well-written manuscript that I think will be quite impactful.

Below, I provide a few technical comments follow by minor editorial suggestions.

**Technical Comments**

+ Figure 2: It would be help to graphically show here the principal stress orientations (and magnitudes) discussed in Lines 139ff. That would help the readers to quickly grasp all tectonic details.

*Reply: We added the stress orientations in the figure as recommended.*

+ Figure 3: For the laymen readers, these 3D plots are hard-to-impossible to put in context. In essence, a detailed map / 3D graphic is needed that shows the locations of these boreholes within the Bedretto Lab. I am not sure if these locations can be easily added to Figure 2, or if another zoomed-in close up near the underground lab is needed. Please consider.

*Reply: We added a clearer 3D view of the borehole setup from two different angles and larger spatial context to Figure 2.*

+ Section 3 (Instrumentation and Experiments) can be and perhaps should be deleted. In my opinion, these details are not needed to understand this study in terms of science, methods, scope and final results. On the other hand, this section distracts from the main "story line" and the main goals of this paper. If deemed important for this completeness purposes, I suggest to move this section into an Esupplement/Appendix.

*Reply: We agree that this section is too long and that it contains information that is irrelevant for the paper. However, deleting it completely is not possible, because its main goal is to describe the succession of the performed experiment that provided the data for the hazard study and also defines the moments, when the hazard study in updated. We believe that this is much clearer now that the section is shortened substantially.*

+ Figure 4: Which magnitude scale is used here, Ml or Mw? Please indicate. In general, since this topic comes up later again, I suggest to explain already early on how magnitudes are estimated, if Ml or Mw is routinely/automatically determined, and with which uncertainties.

*Reply: It is indeed important to clarify early on that the magnitudes reported and used for the analysis are Mw. We added a sentence in Section 3 and added Mw to the axes labels.*

+ Line 275: The "simplifying assumption that the b-value remains constant during injection and after shut-in" is an interesting point to (re-)consider. First of all, is that assumption valid? Given the wealth of data and the experience of the team of author, this should be a very quick and easy point to check and verify. My suspicion is that this is not the case, looking at Figure 4. Perhaps time-dependent b-values, and the variations, over different time-window lengths could be computed to check if/when this assumption is correct. And if not, then we need to think about how this may be propagated into the later hazard calculation.

*Reply: Considering a variable b-value is not possible in our approach, because we do not use a time-dependent induced seismicity model. Thus, any potential variability in the b-value must be accounted for by the aleatory and epistemic uncertainty of the constant b-value used. We agree that the impact of time-dependent b-values on a priori hazard analysis should be investigated. However, this must be subject of future research.*

+ Figure 7: Panels a) and b) need some modifications. First, the y-axis range in both panels should be identical. Second, the grey-scale density plot in panel b) is too fuzzy and doesn't allow being able to see details. I suggest to use a distinct colorbar with 6-10 visually clearly separable colors (say at 0.1, 0.2, 0.3 …) so that details can be seen.

*Reply: We modified the figures as recommended.*

+ Lines 326 - 335: Here, reference should be made to Galis et al (2017) (already in the reference list) and perhaps to Gabriel et al (2024, in Science) and Palgunadi et al (2024, JGR) on arrested and run-away ruptures in complex-geometry fault systems.

*Reply: We agree the referring to the work by Galis et al is appropriate here and we added the reference. Although the work by Gabriel et al and Pagunadi et al do shoot into a similar direction in terms of finding different rupture propagation regimes, they are focussed on interaction between small earthquakes and larger rupture rather than induced earthquakes.*

+ Lines 347 - 352: Using a simple constant stress-drop assumption to translate an estimated fault dimension to a possible event magnitude seems too simplistic, too approximate, and does not include any uncertainty. I strongly recommend to apply modern source-scaling relations (i.e. Thingbaijam et al, 2017), possibly also considering different faulting styles, to estimate potential event magnitude and its range.

*Reply: Comparing our value for the local tectonic Mmax of 5.4 with the scaling relations by Thingbaijam et al, (2017), we find that our estimate is reasonably well in agreement. We added this to the text. We also emphasize stronger that the uncertainties in the scaling relations is generously covered by the standard deviation of 0.8, which not only includes these uncertainties but also those brought in by the estimate of the potential rupture area.*

+ Figure 8: I strongly recommend to plot the scaling relations van der Elst et al (2016) and Galis et al (2017) into this figure for completeness and reference. (This will also shorten the figure caption by two lines …).

*Reply: We added the scaling relations by van der Elst et al and Galis et al to give a more complete picture on the recent research on the Mmax topic.*

+ Section Ground motion models: As someone who has experience in PSHA and GMM's for "standard" regional/national seismic hazard assessment, I am puzzled that PGV is used as ground-motion intensity metric, instead of PGA. I realize this may be the engineering/operational practice in mining-seismicity studies, but this is very counter-intuitive, in particular because we are dealing with very small events that are dominated by high-frequency radiation, and hence PGV may not be an ideal shaking parameter to use. I suggest that the authors provide some clarification and rationale for their choice,.

*Reply: we realize that this is somewhat puzzling because PGA is the more commonly used metric to represent hazard. The main reasons for this decision are:*

- *We wanted to use metrics for TLS thresholds that are in agreement with what the Swiss norm indicate for comparable cases in terms of shaking frequencies. This is the Swiss Norm 640 312a,*

*which deals with constant vibration and occasional vibrations. The thresholds therein are given in PGV.*

- *The damage scenarios deemed most relevant in our study are cracking of tunnel walls and ceiling, rock fall, rock burst etc.. Damage thresholds for these scenarios stem from mining literature and are given in PGV with comparable values to the recommendations of the Swiss Norm.*
- *Most GMMs in literature that deal with our magnitude level and distances come from mining literature and are given in PGV rather than PGA. An example is also for this is also the PSHA study for a mine by Wesseloo (2018).*

*It is not entirely clear, why the relevant literature for our scale and magnitude level deals with PGV rather than PGA. A reason may lie in the fact that traditionally ground velocity is easier to detect with standard devices, because they are more sensitive to the ground motions at this magnitude level.*

*We feel that these aspects should be clarified better in the manuscript and summarize these points at the beginning of this section.*

+ Figure 9: The Cai-Kaiser (2018) model seems to be an almost exact replicate of McGarr & Fletcher (2005), just shifted downwards by "-1 log bias unit". Is that the case? Perhaps an explanatory sentence.

*Reply: We found a small mistake in the script creating this figure. Although the difference between data and the model has decreased a little, there is still a shift between the Cai-Kaiser and the McGarr-Fletcher models. We do not have an explanation for this other than that the fitted parameters in both publications differ because they must have relied on different datasets. We added a sentence clarifying this.*

+ Line 411: The wording "site-specific information" confuses me here, since it is not clear what the "site" is. In BULGG, there are numerous seismic sensors that each could be considered a "recording site". On the other hand, the overall spatial foot-print of BULGG or any similar experimental facility is rather small and would be typically considered as a "single site" in any local/regional PSHA study. Please clarify.

*Reply: We reworded the term site-specific here to be clear that we mean information from the BULGG seismic network.*

+ Line 420: "induced earthquake … have frequencies higher than 10 Hz" —> this relates back to my comment above: Why is then PGV a useful ground-motion metric? And wouldn't PGA make much more sense?

*Reply: see our answer to the earlier comment.*

+ Figure 14: For the 10 panels shown on top, I suggest not to use a continuous colorbar-scale, but one with 8-12 clearly distinct color. Visually, the hues of red between, say 200 - 800 mm/s cannot be discriminated.

*Reply: we changed this according the suggestions by the reviewer.*

+ Line 537-539: The fact that PSHA estimate increase as more data are added is in fact a widely occurring but not well appreciated fact, in general; not only in the context of induced seismicity. I suggest to add corresponding references from the PSHA literature.

*Reply: We added a comment that PSHA suffers from the same "problem" and added the references by Bommer and Abrahamson, (2006) as well as the review paper by Gerstenberger et al (2020).*

+ Sub-section Scale dependent seismogenic response: I would have expected at least a short discussion on whether there are dependencies of the b-value on the faulting-style of the earthquake. Schorlemmer at al (2005?) found a very compelling dependence of the b-value given the faulting-style, which in turn can be explained by the dominant acting stress regime. I suggest to add a few sentences on this here.

*Reply: we added the reference by Schorlemmer et al (2005) as well as Petruccelli et al (2019) and also indicate that Scholz (2015) sees a stress-dependence of b-values.*

+ Line 606: somewhere close to the reference to Deichmann (2017) and in this section there should also be made reference to two papers by Bethmann et al (BSSA, 2011, and GJI, 2012) that examine Mw-Ml scaling relations and site/attenuation effects on Ml/Mw estimates in Switzerland.

*Reply: we added this reference as suggested.*

Editorial Suggestions
* * *
+ The authors refer to "hazard" and "risk" numerous times in the paper, and I do understand that they want to clearly distinguish the two. However, in several instances this distinction is not clear and then things become confusing. Because there are no risk calculations included here and risk is only referred to in a general sense, I suggest the authors add a specific "item at risk" in corresponding statements, for example "risk for tunnel collapse", or "risk to geothermal surface facilities" to better guide the readers what they in mind in each case.

*Reply: we went through the text and became more specific on the term risk or replaced it by hazard if more appropriate in the context.*

+ Please carefully check the punctuation. I noticed many missing periods (" .") to conclude sentences, but even more so I found incorrect setting of commas (" ,") that lead to confusion in terms of meaning of the respective sentences.

*Reply: we did go through the punctuation carefully and replaced a few incorrect commas.*

Other points:

+ Line 100: move "Sweden" after Aspo (in Line 99)

*Done.*

+ Line 106: "intense" is not a good word here, as it cannot be quantified. Use something more specific: real-time; high-resolution (in space, time and frequency frequency) or something like that …

*Done.*

+ Line 109: "seismic risk" … see above …

*Reply: Changed to hazard.*

+ Line 281: (and others) - the text refers to Table 1, but this does not exist; it is Table A1 in the Appendix. This may just be a formatting issue or problem with the latex-template, but please check such referencing carefully.

*Reply: Corrected.*

+ Line 314: The wording "moderate" seems unclear here. Moderate "magnitude"? But what magnitude would that be in the context of the event sizes shown here? Or perhaps better "more frequent events"?

*Reply: we replaced "moderate" by "smaller magnitudes occurring more frequently".*

+ Line 390: The "Table" mentioned here is given in the Appendix. Please correct.

*Reply: in fact we refer to the table 2 in Douglas et al., (2013). We changed this to be more clear.*

+ Line 463: Figure caption to Figure 11: The "hazard" here should be clearly specified as "Hazard to exceed a certain earthquake magnitude". Most readers associate "hazard" with "seismic" (i.e. "shaking hazard") …

*Reply: changed for more clarity.*

+ Line 487: The title to this sub-section should be "seismic" or 'shaking' hazard …

*Reply: changed as suggested.*

+ Line 505: See comment above to Line 463 / 483

*Reply: changed as suggested.*

+ Lines 540 - 544: This sentence seems garbled up; I cannot understand it. Also, change "became" to "become".

*Reply: Corrected and split in two sentences for more clarity.*

+ Line 548: remove or quantify "somewhat"

*Reply: Removed*

+ Line 596: abbreviation "GSK" not defined$

*Reply: We define GSK at first use.*

---

## Author Comment (AC2)

**Reponses to the reviewer's comment 2 on manuscript egusphere-2024-3882**

**Review by Mauro Cacace, 04 Apr 2025**

*Dear Mauro Cacace*

*Thank you very much for the thorough review and the many useful comments and suggestion. They are addressed as stated in the responses to you comment below and have led to an improvement of the manuscript on our study.*

**Overall comment:** The study by Gischig and co-authors present a review on lessons learnt from hectometer-scale stimulations done in the Bedretto Underground Laboratory on the feasibility/merit/limitations/open challenges for probabilistic seismic hazard estimates during hydraulic stimulation. The authors present their workflow to what they referred to a PISHA, which includes data collected and available at different stages, as derived from other geothermal projects and those more specific to their underground laboratory during past projects. In their workflow, data are used to update at each time the computed (in a probabilistic sense) seismic hazard, which they cross-correlate mainly to operational parameters (injected fluid volume), and, relative in a weak manner to the local geology/tectonic. The authors consider an additional layer to better refine their PISHA by including GGMs and discuss in the final chapter of the study the benefits of their multi-component and "time-dependent" workflow in the light of existing (A)TLSs.

I personally found the manuscript scientifically sound and well organized, from the introduction to the problem, associated open question(s), proposed solution(s) --> data/modelling/results and implications/next steps. While the manuscript is in general well written, there are some parts where the authors could (and should) improve the level of details in order to ease the efforts from the readers to not only completely follow their procedure but also to properly judge the scientific merit of each step described. On similar lines, while I agree with the authors' choice on the final discussion points, I personally found all 3 sub-paragraph filled with too many generic statements and I would advise the authors to carefully reconsider those by adding concrete explanations to their sentencing.

I'm listing some (minor) open questions/suggestions to improve the readability/clarity and sometimes the scientific output of the manuscript (considering what has been already discussed in the previous post by the other reviewer), which I consider fits well with the topic of SE and would make a nice contribution to the journal.

**Reply:** *We appreciate the overall assessment of the manuscript and, in particular, agree that the discussion section is somewhat too extensive and contains statements that do not add much to the main messages and may be omitted. We went through the discussion and removed some statements that we feel made the text more difficult to read rather than help to general understanding. Together with the suggestions for improvement by Review 1, we feel that this has led to an improvement.*

\* Abstract (line 40-41): While I agree that a different seismogenic response between deep reservoir studies and underground laboratory is likely to be related to specific differences in their settings (stress levels and fault area) as well as in the operations (injected volume), I have some difficulties in how this information can be used to properly (i.e. in a quantitative manner)

used to propose/advance safer exploitation concepts. After reading through the whole of the manuscript I was expecting a discussion point addressing this specific issue, but the authors failed to take it up later in the paper. This said, I would consider either to avoid such generic sentences or at least rephrase them to read less abstract and more scientifically enriched.

*Reply: The statement was meant as an outlook towards future research and not as an outcome of our study. We agree that such statements are more appropriate in the discussion and decided to omit it in the abstract.*

\* Abstract (concluding sentence): A first-order control here stems from the local geology and geological knowledge that is orders of magnitude simpler/known/understood in underground laboratories than in the field. In addition, also controlled conditions of an underground laboratory are hard to achieve in the field. All these aspects contributes as the authors stated in a "safer seismic hazard", but also makes the "up-scaling" of the applications hard.

*Reply: We agree with this comment and added a statement onf the limitation regarding upscaling to the concluding sentence.*

\* Introduction (line 58-60): while discussing real forecasting, a bit of caution here. To my knowledge there is no approach we can rely upon to forecast induced seismic hazard. What current approaches offer is to either statistically project in time previous knowledge (as in this study) or at best hindcast (with diverse success) induced seismic hazard.

*Reply: We fully agree that this should be written cautiously. We reworded the statement and added that no reliable forecast models exist.*

  \* Introduction (lines 70 onward): The authors should add that thresholds in (A)TLS are likely to be empirically derived (based on experts knowledge and/or previous experience) and should potentially also be considered as an additional source of (potentially epistemic) uncertainties in PSHA (which they are not).

*Reply: We added a sentence highlighting that TLS thresholds are indeed mostly determined by expert judgment.*

\* Introduction (lines 75/76 onward): while discussing Mmax, please review the study by van der Elst and co-workers (https://doi.org/10.1002/2016JB012818), where the authors nicely showcased that whether it is true that the Mmax can scale with injected (net) volume (in reality they should rather scale with the previous earthquake population) there is only poor (if not at all) control their exact position in the seismicity population, that is, Mmax occurrence can be at best randomly picked within the statistics. This poses some questions on the feasibility of TLS thresholds, as demonstrated for real field applications by post-injection seismicity, which "hosts" preferentially the largest magnitude seismic event (lessons learnt from Pohang, Vendenheim, Soultz and many others).

*Reply: We added the statement referring to the work of van der Elst and on its implications on the seismicity trailing effect.*

\* Introduction (line 95/97): This is an excellent question, I like it a lot. Caveat here: how to cast the governing physics (only partially known/understood) into a probabilistic approach? The same is true to a certain degree for underground laboratories, which target a specific fault of a

limited extent under controlled conditions that are really hard to achieve in any "real" field application.

*Reply: Thank you for this comment. We generally agree with the comment. However, at the scale of the Bedretto underground laboratory several faults and fracture system could be targeted so that together with the overall scale, we are closer to the real field application.*

* Method (line 249/251): mean and median are not the same thing, and they provide different outcomes. In addition, stating that "Conservatism" comes from a conservative choice reads at least redundant. Please, consider rephrasing this sentence in order to clarify the message (also by considering that the choice of the traffic light system is empirical it not subjective to the experts' knowledge).

*Reply: Rhe two sentences were reworded for more clarity.*

* Method - Magnitude rates (eq 1):

 - V(t) should rather be V_dot(t) (during injection)

*Reply: We corrected this error.*

 - This is more about personal taste. I have some hard times to understand the main idea behind the post shut-in definition of the seismicty rate (from the original paper). As a matter of fact the equation shows (as it should given observation) the same traits of a typical exponential decaying (not too much dissimlar to an Omori law), but it has apriori parameters (e.g. V_dot(t_shut-in) and tau) that are way harder to constraint than more classical approaches based on a (modified) Omori Law. As an example I find it difficult to have it representing any tailing in time if not by correlating injection rate at shut-in to the corresponding overpressure computed/monitored (this also assumes linearity in the pressure reservoir response which is not always the case).

*Reply: Clearly, there are different ways to express trailing seismicity. For the reasons mentioned by the reviewer, we chose a much simpler approach and represent trailing seismicity with the percentage of seismic events that occurred after shut-in. The simplicity and the fact, that we are not interested in the time-dependent seismicity after shut-in but the total seismicity, justifies in our view this approach.*

-   Any  explanation  behind  the  reference  (0.05)  b-value?
 - Same as above for the 10% of post shut-in seismicity?

*Reply: These values were chosen heuristically and roughly match a median value of those cases studies for which these values are reported. We indicate this in the text.*

* Method - maximum moment magnitude  (line 310) - how physical considerations come into play here?

*Reply: This is described in the paragraphs that follow. We reworded the sentence to make this clear.*

* Method - maximum moment magnitude (line 312-314) - honestly speaking this sentence/remark is not true (or at least not always), see the recent seismicity at Vendenheim project.

*Reply: This is a misconception of our statement. While the maximum expected and/or observed magnitude may indeed have an impact on a project in terms of the associated risk, the physical upper bound of induced earthquakes is a very rare event the would produce substantial damage to a project or the surrounding infrastructure, but occurs at such a low rate that the risk (probability of damage to occur) is very low. The reference in the text clarify this issue.*

* Method - maximum moment magnitude (line 322-325) - A bit of caution here, lessons learnt form Pohang entails a tectonic control on Mmax as per classical theory.

*Reply: We fully agree – and also state it in our text – that cases like Pohang show that Mmax has been controlled by what is tectonically possible. However, this may not always be the case depending on depth, stress regime, and orientation and state of faults accessed by the high-pressure fluid injection as many authors argue. Our choices of Mmax tries to reflect these different views and outcomes.*

* Method - maximum moment magnitude (line 341-343) - Please refer also o the study by Galis et al (2017 - DOI: 10.1126/sciadv.aap7528) on exactly this topic.

*Reply: We added the study by Galis et al to the discussion on Mmax and also included the limits in Figure 8 as also proposed by reviewer 1. We do not consider these limits in the non-tectonic volume-dependent choice of Mmax, because the McGarr limit already covers this option sufficiently well.*

* Method - maximum moment magnitude (line 350-351) - from where the 3 MPa stress drop comes?

*Reply: The value is an average value that can be seen as representative across many magnitude levels. We added a statement with reference to Cocco et al (2016) in the manuscript.*

* Results - Magnitude rates
- It's not clear, and I have my limitation to it, why the authors don't discuss normalized PDFs for the exceedance probability. I warmly advise the authors to add their own point of view/explanation, given that all their results read to a certain level "biased" by this choice.

*Reply: The figure 11 presents the probabilitiy of exceeding a certain magnitude and at the same time presents the uncertainty in these estimates as grey shading. We do not fully understand what the reviewer means with normalized PDFs in this context.*

- While discussing GMMs, the "unreasonable" range might stem from the high frequency content (see the comment from the previous reviewer)

*Reply: We highlight that the actual reason for this may be related to combining several GMMs leading to added epistemic and aleatory uncertainties.*

* Discussion - sensitivities and uncertainties

Introducing the discussion paragraph with a rather generic sentence of benefits from PISHA should be followed by a detailed discussion of what those benefits are. I missed this. In addition, sometimes the authors state the obvious as while discussing the median and percentile sensitivity (percentiles provide a view of the data distribution)

*Reply*: *We rewrote part of the section to bring across the main message of it (which is relative sensitivity of uncertainties to GMM, seismogenic properties and Mmax) more concisely.*

\* Discussion - scale and depth dependent seismogenic response

Again here the authors discusses aspects that have been already discussed/proposed in previous study and that, to my own reading of their manuscript, are not completely related to what was presented. Their concluding sentence reads too generic. It is not clear how studies based on underground laboratories help in addressing the problems described above. Please note that I do agree that such studies are extremely important, and this is why I would advise the authors to discuss what in their opinions are opportunities from those studies as it would greatly advance the scientific merit of the discussion.

*Reply*: *We feel that the discussion prior to the concluding sentence explains how we reach this conclusion. If we understand the reasons why some geological, hydromechanical or operational aspects lead to weaker responses, we may find ways to reduce induced seismicity in full-scale operations. We reworded the sentence to be more specific on this.*